# Spatiotemporal control of cell cycle acceleration during axolotl spinal cord regeneration

**Emanuel Cura Costa[1†], Leo Otsuki[2†], Aida Rodrigo Albors[3], Elly M Tanaka[2], Osvaldo Chara[1,4*]**

[1]Systems Biology Group (SysBio), Institute of Physics of Liquids and Biological Systems (IFLySIB), National Scientific and Technical Research Council (CONICET) and University of La Plata (UNLP), La Plata, Argentina; [2]The Research Institute of Molecular Pathology (IMP), Vienna Biocenter (VBC), Vienna, Austria; [3]Division of Cell and Developmental Biology, School of Life Sciences, University of Dundee, Dundee, United Kingdom; [4]Center for Information Services and High Performance Computing, Technische Universität Dresden, Dresden, Germany

**Abstract:** Axolotls are uniquely able to resolve spinal cord injuries, but little is known about the mechanisms underlying spinal cord regeneration. We previously found that tail amputation leads to reactivation of a developmental-like program in spinal cord ependymal cells (Rodrigo Albors et al., 2015), characterized by a high-proliferation zone emerging 4 days post-amputation (Rost et al., 2016). What underlies this spatiotemporal pattern of cell proliferation, however, remained unknown. Here, we use modeling, tightly linked to experimental data, to demonstrate that this regenerative response is consistent with a signal that recruits ependymal cells during ~85 hours after amputation within ~830 μm of the injury. We adapted Fluorescent Ubiquitination-based Cell Cycle Indicator (FUCCI) technology to axolotls (AxFUCCI) to visualize cell cycles in vivo. AxFUCCI axolotls confirmed the predicted appearance time and size of the injury-induced recruitment zone and revealed cell cycle synchrony between ependymal cells. Our modeling and imaging move us closer to understanding *bona fide* spinal cord regeneration.

**\*For correspondence:**
osvaldo.chara@tu-dresden.de

[†]These authors contributed equally to this work

**Competing interest:** The authors declare that no competing interests exist.

## Introduction

The axolotl (*Ambystoma mexicanum*) has the remarkable ability to regenerate the injured spinal cord (reviewed in *Freitas et al., 2019*; *Tazaki et al., 2017*; *Chernoff et al., 2003*), and thus represents a unique system to study the mechanisms of successful spinal cord regeneration. Key players in this process are the ependymal cells lining the central canal of the spinal cord, which retain neural stem cell potential throughout life (*Becker et al., 2018*).

In earlier studies, we found that spinal cord injury in the axolotl triggers the reactivation of a developmental-like program in ependymal cells, including a switch from slow, neurogenic to fast, proliferative cell divisions (*Rodrigo Albors et al., 2015*). We showed that in the uninjured spinal cord and in the non-regenerating region of the injured spinal cord, ependymal cells divide slowly, completing a cell cycle in 14.2 ± 1.3 days. In contrast, regenerating ependymal cells speed up their cell cycle and divide every 4.9 ± 0.4 days (*Rodrigo Albors et al., 2015*; *Rost et al., 2016*). By using a mathematical modeling approach, we demonstrated that the acceleration of the cell cycle is the major driver of regenerative spinal cord outgrowth and that other processes such as cell influx, cell rearrangements, and neural stem cell activation from quiescence play smaller roles (*Rost et al., 2016*). We quantitatively analyzed cell proliferation in space and time and identified a high-proliferation

zone that emerges 4 days after amputation within the 800 μm adjacent to the injury site and shifts posteriorly over time as the regenerating spinal cord grows (*Rost et al., 2016*). In particular, we quantified a switchpoint separating the high-proliferation from the low-proliferation zones from day 4 on (*Figure 1—figure supplement 1*). What underlies this precise spatiotemporal pattern of cell proliferation in the regenerating axolotl spinal cord, however, remains unknown. Pattern formation phenomena occurring during development can be quantitatively reproduced by invoking morpho-genetic signals spreading from localized sources (*Morelli et al., 2012*). It is thus conceivable that tail amputation triggers a signal that propagates or diffuses along the injured spinal cord to speed up the cell cycle of resident cells.

In this new study, we take a modeling approach supported by previous and new experimental data to unveil the spatiotemporal distribution that such a signal should have in order to explain the observed rate of spinal cord outgrowth in the axolotl. We confirm several of our theoretical predictions by generating a new transgenic AxFUCCI axolotl that faithfully reports cell cycle phases in vivo using axolotl-specific cell cycle protein fragments. We envision that AxFUCCI axolotls will serve as useful tools for future studies of proliferation during development and regeneration. Together, our results provide new clues for when and where to search for the signal/s that may be responsible for driving successful spinal cord regeneration.

## Results
### Model of uninjured spinal cord
Taking into account the symmetry of the ependymal tube and that ependymal cells organize as a pseudo-stratified epithelium (*Joven and Simon, 2018*), we modeled the anterior-posterior (AP) axis of the spinal cord as a row of ependymal cells (see Section 1.1 for more details). We modeled ependymal cells as rigid spheres of uniform diameter and assumed that they can be either cycling or quiescent and defined the fraction of cycling cells as the growth fraction, *GF*. We modeled the proliferation dynamics of cycling cells as follows: we assumed that in the initial condition each cycling cell is in a random coordinate along its cell cycle, where the initial cell cycle coordinate and the cell cycle length follow an exponential and a lognormal distribution, respectively. In the uninjured axolotl spinal cord, upon cell division, (i) the daughter cells inherit the cell cycle length from the mother's lognormal distri-bution and (ii) the daughter cells translocate posteriorly, displacing the cells posterior to them. This last feature of the model is the implementation of what we earlier defined as 'cell pushing mechanism' (*Rost et al., 2016*). This model predicts that after a time of approximately one cell cycle length mitotic events will occur along the AP axis and contribute to the growth of the spinal cord (*Figure 1A*).

### Model of regenerating spinal cord
Next, we removed the most posterior cells of the tissue to model tail amputation and the regenera-tive response in the remaining cells ($N_0$) in silico (see *Figure 1B* and Section 1.1 for more details). We assumed that amputation triggers the release of a signal that spreads anteriorly from the injury site with constant speed along the AP axis, recruiting cells by inducing a change in their cell cycle. We established that cell recruitment stops at time $\tau$, recruiting $\lambda$ μm of cells anterior to the amputation plane. We notated the AP position of the most anterior cell recruited by the signal as $\xi(t)$ and called this the recruitment limit, such that $\xi(t = \tau) = -\lambda$. In the model, all cycling cells anterior to the cell located at $\xi(t)$ are not recruited and continue cycling slowly during the simulations (*Figure 1—figure supplement 2*). In contrast, cycling cells posterior to $\xi(t)$ are recruited at a time $t$ within the interval $0 \leq t \leq \tau$ and irreversibly modify their cycling according to their cell cycle phase at the time of recruitment.

Because we previously demonstrated that the length of $G_2$ and M phases does not change upon amputation (*Rodrigo Albors et al., 2015*), we assumed that cells in $G_2$ or M within the recruitment zone spend the same time to divide as the non-recruited cells. However, their daughters will reduce their cell cycle (see below) (*Figure 1C*, Section 1.1.3). In contrast, because we showed that regener-ating cells go through shorter $G_1$ and S phases than non-regenerative cells ($G_1$ shortens from 152 ± 54 hours to 22 ± 19 hours; S shortens from 179 ± 21 hours to 88 ± 9 hours, *Rodrigo Albors et al., 2015*), we reasoned that the signal instructs recruited cells to shorten $G_1$ and S, effectively shortening their cell cycle.

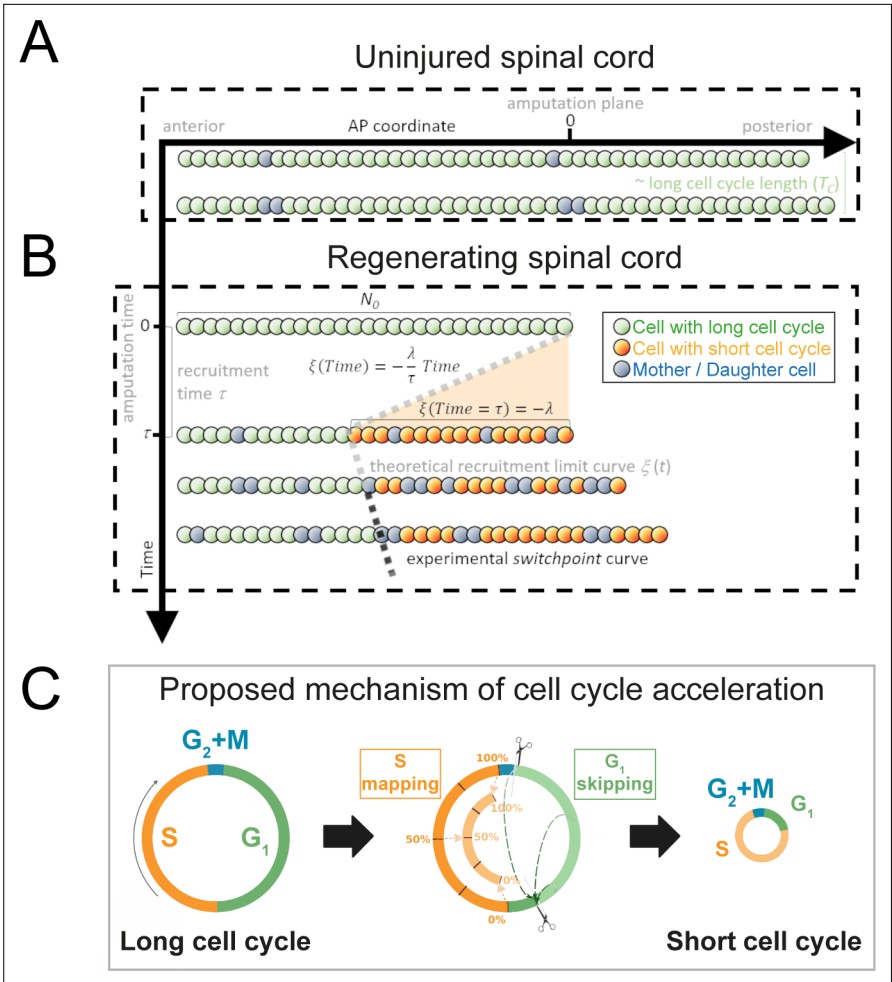

**Figure 1.** Model of uninjured and regenerating spinal cord growth based on shortening $G_1$ and S phases. (**A**) Uninjured spinal cord. 1D model of ependymal cells aligned along the anterior-posterior (AP) axis. In uninjured tissue, ependymal cells cycle with a long cell cycle length (see ***Figure 1—figure supplement 1***). When they divide, they 'push' cells posteriorly and the spinal cord tissue grows. As an example, two mother cells (in blue) in the top row give rise to four daughter cells in the second row (also in blue) following one division. This results in a growth of two cell diameters within a timeframe of approximately one long cell cycle length. (**B**) Regenerating spinal cord. After amputation (AP coordinate and time equal to zero), a signal is released anteriorly from the amputation plane during a time $\tau$ and spreads while recruiting resident ependymal cells up to the theoretical recruitment limit $\xi$ located at $-\lambda$ μm from the amputation plane. After a certain time, the recruitment limit $\xi$ overlaps the experimental switchpoint (see definition of the experimental switchpoint in ***Figure 1—figure supplement 1***). (**C**) Proposed mechanism of cell cycle acceleration, consisting of partial skipping of $G_1$ phase and proportional mapping between long and short S phases. In the middle panel, two examples are depicted (dashed green arrows) of cells that were in the long $G_1$ phase (immediately before recruitment) that become synchronized (immediately after recruitment). Additionally, there are three examples (dotted orange arrows) of cells that were in the long S phase (immediately before recruitment) that are proportionally mapped (immediately after recruitment). All these examples are shown in detail in ***Figure 1—figure supplement 2***. The diameter of the circles is approximately proportional to the length of the cell's cell cycle.

The online version of this article includes the following figure supplement(s) for figure 1:

**Figure supplement 1.** Space-time distribution of cell proliferation during axolotl spinal cord regeneration.

**Figure supplement 2.** Further explanation of the cell cycle coordinate transformation invoked by the model.

To explain how ependymal cells may shorten $G_1$ and S phases in response to the injury signal, we conceived a mechanism of $G_1$ shortening in which a certain part of this cell cycle phase is skipped. We implemented this mechanism as follows (**Figure 1C**, **Figure 1—figure supplement 2**, Section 1.1.1): if at the time of recruitment the cell is in $G_1$, there are two possible coordinate transformations. If the cell cycle coordinate is located before the difference between the long (non-regenerative) $G_1$ length and the shortened (regenerative) $G_1$ length, the cell clock is reset; that is, its transformed cell cycle coordinate will be the beginning of the shortened $G_1$ phase in the next simulation step (immediately after $G_2$+ M phases). In contrast, if the original cell cycle coordinate of the cell is located after the difference between the long $G_1$ length and the shortened $G_1$ length, there is no change in the time to enter into S phase. The difference between the long and short $G_1$ lengths constitutes a temporal threshold. If the $G_1$ cell lies before this value, it skips, and if it lies after, it continues cycling as before. This mechanism of $G_1$ skipping predicts a partial synchronization of the cell cycle as cells transit through $G_1$ (**Figure 1— figure supplement 2**) – an important implication that we test experimentally later.

Because all DNA must be duplicated prior to cell division, we considered a different mechanism to model S phase shortening: if the cell cycle coordinate belongs to S at the moment of recruitment, the new cell cycle coordinate of this cell will be proportionally mapped to the corresponding coordinate of a reduced S phase in the next simulation step (**Figure 1C**, **Figure 1—figure supplement 2**, Section 1.1.2). For instance, if a cell is 40% into its long S phase when the recruitment signal arrives, it will be 40% into its shorter S phase in the next simulation step.

Daughter cells of recruited cells inherit short $G_1$ and S phases from their mothers and consequently have shorter cell cycle lengths (**Figure 1C**). In particular, daughter cells whose mother was in $G_2$ + M phases at the moment of recruitment will transit through short $G_1$ phases, effectively adopting a

**Table 1.** Model parameters.

| Model parameter | Value/explanation | Fixed/free |
| --- | --- | --- |
| $G_1$ phase non-regenerating mean | 152 hours | |
| $G_1$ phase non-regenerating sigma | 54 hours | |
| S phase non-regenerating mean | 179 hours | |
| S phase non-regenerating sigma | 21 hours | |
| $G_2$ + M phases non-regenerating mean | 9 hours | |
| $G_2$ + M phases non-regenerating sigma | 6 hours | |
| $G_1$ phase regenerating mean | 22 hours | |
| $G_1$ phase regenerating sigma | 19 hours | |
| S phase regenerating mean | 88 hours | |
| S phase regenerating sigma | 9 hours | |
| $G_2$ + M phases regenerating mean | 9 hours | |
| $G_2$ + M phases regenerating sigma | 2 hours | |
| *GF* non-regenerating | 0.12 | |
| Cell length along the AP axis | 13.2 µm | |
| $t_{G0\text{-}G1}$ | 48 hours | Fixed parameters, extracted from ***Rodrigo Albors et al., 2015*** |
| $N_0$ | Initial number of cells along the AP axis, anterior to the amputation plane | |
| $\lambda$ | Maximal length from the amputation plane recruited by the signal (µm) | |
| $\tau$ | Maximal time for cell recruitment (days after amputation or hours after amputation) | Free parameters (determined in this study) |

GF: growth fraction; AP: anterior-posterior.

short cell cycle length. Finally, we assumed that recruitment of a quiescent cell would induce its progress from $G_0$ to $G_1$ after an arbitrary delay (Section 1.1.4). To parametrize the cell phase durations of recruited and non-recruited cycling cells, the growth fraction, and cell geometry, we used our previous experimental data from regenerating and non-regenerating regions of axolotl spinal cords (*Rodrigo Albors et al., 2015*) (Section 1.2 and *Table 1*).

The model predicts that if we wait a time similar to the short cell cycle length, we will observe more proliferation posterior to $\xi$ than anterior to it. In particular, if this model is correct, the prediction for $\xi$ (*Figure 1B*) should agree with the experimental curve of the switchpoint separating the region of high proliferation from low proliferation along the AP axis of the axolotl spinal cord during regeneration (see definition of the switchpoint curve in *Figure 1—figure supplement 1*).

## Regenerative spinal cord outgrowth can be explained by a signal that acts during the 85 ± 12 hours following amputation and recruits cells within 828 ± 30 μm of the amputation plane

To evaluate if the model could explain the regenerative outgrowth of the ependymal tube and to estimate the free parameters, we fitted $\xi(t)$ to the experimental switchpoint (*Rost et al., 2016*). Specifically, we followed an inference Approximate Bayesian Computation (ABC) method, streamlined by the use of pyABC-framework (*Klinger et al., 2018*). The model successfully reproduced the experimental switchpoint with best-fitting parameters $N_0$ = 196 ± 2 cells, $\lambda$ = 828 ± 30 μm, and $\tau$ = 85 ± 12 hours (*Figure 2A*; parameter posterior distributions obtained after convergence are shown in *Figure 2—figure supplement 1*, and see Section 1.3 for details). Interestingly, a clonal analysis of the model shows that while the anterior-most cells are slightly displaced, cells located close to the amputation plane end up at the posterior end of the regenerated spinal cord (*Figure 2—figure supplement 2A*), in agreement with cell trajectories observed during axolotl spinal cord regeneration (*Rost et al., 2016*). Additionally, the velocity of a clone monotonically increases with its AP coordinate (*Figure 2—figure supplement 2B*), also in line with experimental data (*Rost et al., 2016*). These results suggest that cells preserve their relative position along the AP axis. Hence, when plotting the relative position of each clone to the outgrowth of the corresponding tissue minus the recruitment limit $\xi(t)$, we observed that this normalized quantity is conserved in time, a fingerprint of the scaling behavior characteristic of regeneration (*Figure 2—figure supplement 2C*). Importantly, with the parameterization leading to the best fitting of the experimental switchpoint, we quantitatively predicted the time evolution of the regenerative outgrowth that was observed in vivo (*Rost et al., 2016*; *Figure 2B*, *Video 1*).

Our model assumed that cells are rigid spheres of uniform diameter fixed from the mean length of ependymal cells measured along the AP axis (*Rost et al., 2016*). To test whether this *naïve* assumption could impact on our prediction of the regenerative spinal cord outgrowth, we repeated the simulations but replacing the mean cell length by the biggest and smallest possible cell lengths within a 99% confidence interval based on earlier data (*Rost et al., 2016*). Spinal cord outgrowth predicted under these two extreme scenarios could hardly be distinguished from the previous prediction (*Figure 2—figure supplement 3A,B*). Similar results were obtained when we assumed that ependymal cells do not have a constant length along the AP axis but one extracted from a normal distribution parametrized from the experimental data on ependymal cell lengths along the AP axis (*Figure 2—figure supplement 3C*).

When we simulated a fast recruitment process by reducing the maximal recruitment time $\tau$ to 1 day while maintaining the maximal recruitment length $\lambda$ constant, we found that the model-predicted outgrowth overestimates the experimental outgrowth (*Figure 2—figure supplement 4A*). On the contrary, when we decreased recruitment speed by increasing $\tau$ to 8 days while keeping $\lambda$ constant, we observed a shorter outgrowth than that observed experimentally (*Figure 2—figure supplement 4A*). Reducing the maximal recruitment distance $\lambda$ to zero mimics a hypothetical case in which the signal is incapable of recruiting the cells anterior to the amputation plane (*Figure 2—figure supplement 4B*). Increasing $\lambda$ by approximately 100% without changing $\tau$ (i.e., increasing recruitment speed) results in more recruited ependymal cells and faster spinal cord outgrowth than observed in vivo (*Figure 2—figure supplement 4B*). These results point to a spatially and temporarily precise cell recruitment mechanism underlying the tissue growth response during axolotl spinal cord regeneration.

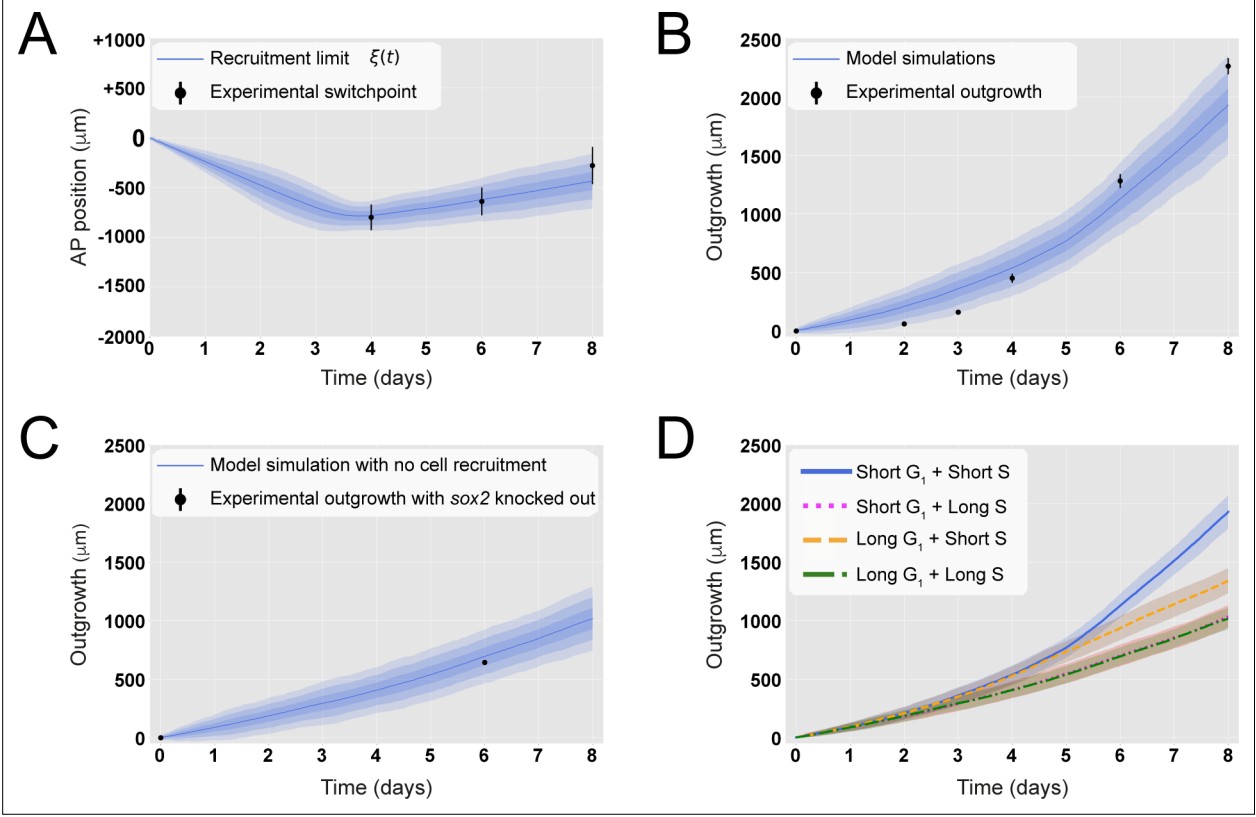

**Figure 2.** A hypothetical signal recruits ependymal cells up to 828 ± 30 μm anterior to the amputation plane during the 85 ± 12 hours following amputation, recapitulating in vivo spinal cord regenerative outgrowth. (**A**) The modeled recruitment limit successfully fits the experimental switchpoint curve. Best-fitting simulations of the model-predicted recruitment limit ξ(**t**) overlap the experimental switchpoint curve (see definition of the experimental switchpoint in *Figure 1—figure supplement 1*). Best-fitting parameters are $N_0$ = 196 ± 2 cells, λ = 828 ± 30 μm, and τ = 85 ± 12 hours. (**B**) The model quantitatively matches experimental axolotl spinal cord outgrowth kinetics (*Rost et al., 2016*). (**C**) The model reproduces experimental outgrowth reduction when the acceleration of cell proliferation is impeded. Prediction of the model assuming that neither S nor $G_1$ phase lengths were shortened superimposed with experimental outgrowth kinetics in which acceleration of the cell cycle was prevented by knocking out *Sox2* (*Fei et al., 2014*). Lines in (**A–C**) show the means while blue shaded areas correspond to 68, 95%, and 99.7% confidence intervals, from darker to lighter, calculated from the 1000 best-fitting simulations. (**D**) S phase shortening dominates cell cycle acceleration during axolotl spinal cord regeneration. Spinal cord outgrowth kinetics predicted by the model assuming shortening of S and $G_1$ phases (blue line), only shortening of S phase (orange dashed line), only shortening of $G_1$ phase (magenta dotted line), and neither S nor $G_1$ shortening (green line). The magenta line and green line are overlapped with one another. Means are represented as lines, and each shaded area corresponds to 1 sigma out of 1000 simulations.

The online version of this article includes the following figure supplement(s) for figure 2:

**Figure supplement 1.** Approximate Bayesian Computation (ABC) fitting of the recruitment limit model to the experimental switchpoint data.

**Figure supplement 2.** The model encompasses the cell pushing mechanism: the more posterior a cell is, the faster it moves.

**Figure supplement 3.** Incorporating variability of cell length along the anterior-posterior (AP) axis does not impact on predicted axolotl spinal cord outgrowth.

**Figure supplement 4.** The maximal recruitment time (τ) and the maximal recruitment length (λ) determine the outgrowth of the axolotl spinal cord.

## Regenerating spinal cord outgrowth when cell cycle acceleration is impeded

We next asked how much the spinal cord would grow if the cell cycle acceleration instructed by the injury signal is blocked. We made use of our model and predicted the tissue outgrowth when the lengths of $G_1$ and S were unaltered after amputation. In this condition, all cells would divide with the durations of cell cycle phases reported under non-regenerating conditions (*Rodrigo Albors et al., 2015*). Our results show that blocking recruitment, and therefore the acceleration of the cell cycle, slows down tissue growth, leading to an outgrowth of 694 ± 77 μm instead of the observed 1127 ± 103 μm at day 6 (*Figure 2C*). This result is consistent with reducing down to zero the maximal recruitment length λ

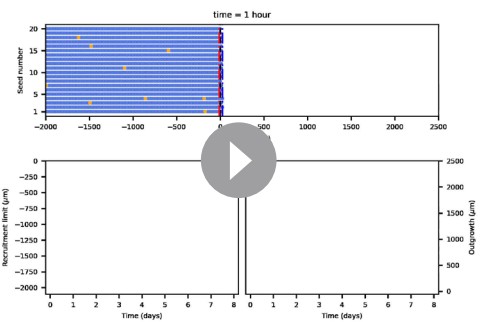

**Video 1.** 1D model simulations of spinal cord regeneration. (Top panel) 20 model simulations from 20 different random seeds (the seeds are shown in the vertical axis). The color code corresponds to the generation of each cell (blue, orange and black correspond to the first, second and third generation, respectively). Vertical interrupted black line denotes the amputation plane (AP coordinate of 0). The recruitment limit $\xi(t)$ (vertical red dashed line) propagates linearly in time up until the time $t$ of 85 hours post amputation, covering up the maximal recruitment $\lambda$ of 828 mm anterior to the amputation plane. (Bottom left panel) Predicted recruitment limit $\xi(t)$ as a function of time from the simulations showed in the top panel. Mean value is depicted in the red line while the red shaded areas corresponds to 68, 95 and 99.7 % confidence intervals (Bottom right panel) Predicted spinal cord outgrowth predicted by the model from the simulations showed in the top paneladapte. The line represents to the mean (also indicated as the blue vertical line in the top panel) and the blue shaded areas correspond to the 68, 95 and 99.7 % confidence intervals. The 20 simulations have the same parametrization than the 1,000 simulations showed in **Figure 2A and B**.

https://elifesciences.org/articles/55665/figures#video1

(**Figure 2—figure supplement 4B**). Interestingly, this model-predicted outgrowth is in agreement with the reported experimental outgrowth in *Sox2* knock-out axolotls, in which the acceleration of the cell cycle does not take place after amputation (**Figure 2C**, **Fei et al., 2014**).

## S phase shortening is sufficient to explain the initial regenerative spinal cord outgrowth

The relative contributions of $G_1$ versus S phase shortening to spinal cord outgrowth are an important unknown that is technically difficult to interrogate in vivo. We made use of our model to address this question in silico. For this, we maintained the same parametrization recapitulating spinal cord outgrowth (**Figure 2A and B**) but modified the model such that recruited cycling cells shorten S phase but not $G_1$ phase (i.e., leaving unaltered $G_1$ phase) or vice versa. Interestingly, our results indicate that shortening of only S phase can explain the explosive spinal cord outgrowth observed in vivo, independently of $G_1$ shortening, up to day 4 (**Figure 2D**, blue line and orange line). In contrast, shortening of only $G_1$ phase has a mild impact on the initial outgrowth as it results in an outgrowth almost identical to the case in which neither $G_1$ nor S phase was reduced (**Figure 2D**, magenta line versus green line). From day 4, though, shortening of only S phase cannot recapitulate the observed outgrowth (**Figure 2D**, blue line and orange line), and indeed, it is the shortening of both S and $G_1$ phases that returns the same outgrowth than that observed in vivo. These modeling predictions are a consequence of (i) the proximity of S phase to the next cell division compared with $G_1$ phase; (ii) the fact that S phase represents ~7.5 days of the total ~14 days of the long cell cycle, which is reduced to ~3.7 days in the ~5 days short cell cycle; and (iii) the time window of the investigated outgrowth being 8 days. To conclude, these results indicate that, up to day 4, shortening of S phase can explain the regenerative spinal cord outgrowth in the axolotl, while the effect of $G_1$ shortening manifests from day 5.

## Visualizing cell cycle progression in axolotls in vivo using FUCCI

Our model makes defined assumptions on how the phases of the cell cycle shorten to result in an acceleration of cell cycle over 85 hours within 828 μm of the injury site. We sought to validate the model by determining the kinetics of this response rigorously using a tool that distinguishes cell cycle phases in vivo while preserving spatiotemporal context. For this, we adapted Fluorescent Ubiquitination-based Cell Cycle Indicator (FUCCI) technology to axolotls (**Figure 3A**; **Zielke and Edgar, 2015**). FUCCI is a genetically encoded reporter that distinguishes cell cycle phases by capitalizing on the mutually exclusive, oscillatory activity of two ubiquitin ligases (**Sakaue-Sawano et al., 2008**). SCF[Skp2] is active in S and $G_2$ of the cell cycle when it targets the DNA licensing factor Cdt1 for proteolytic degradation. In contrast, APC/C[Cdh1] is active from mid-M to $G_1$; during these phases, it targets Geminin (Gmnn; a Cdt1 inhibitor) for degradation. Fusing the degradation-targeting motifs (degrons) in the Cdt1 and Gmnn proteins to two distinct fluorophores puts fluorophore abundance under the control of SCF[Skp2] and APC/C[Cdh1] activity and enables fluorescence to be used as a readout for cell cycle phase. Importantly,

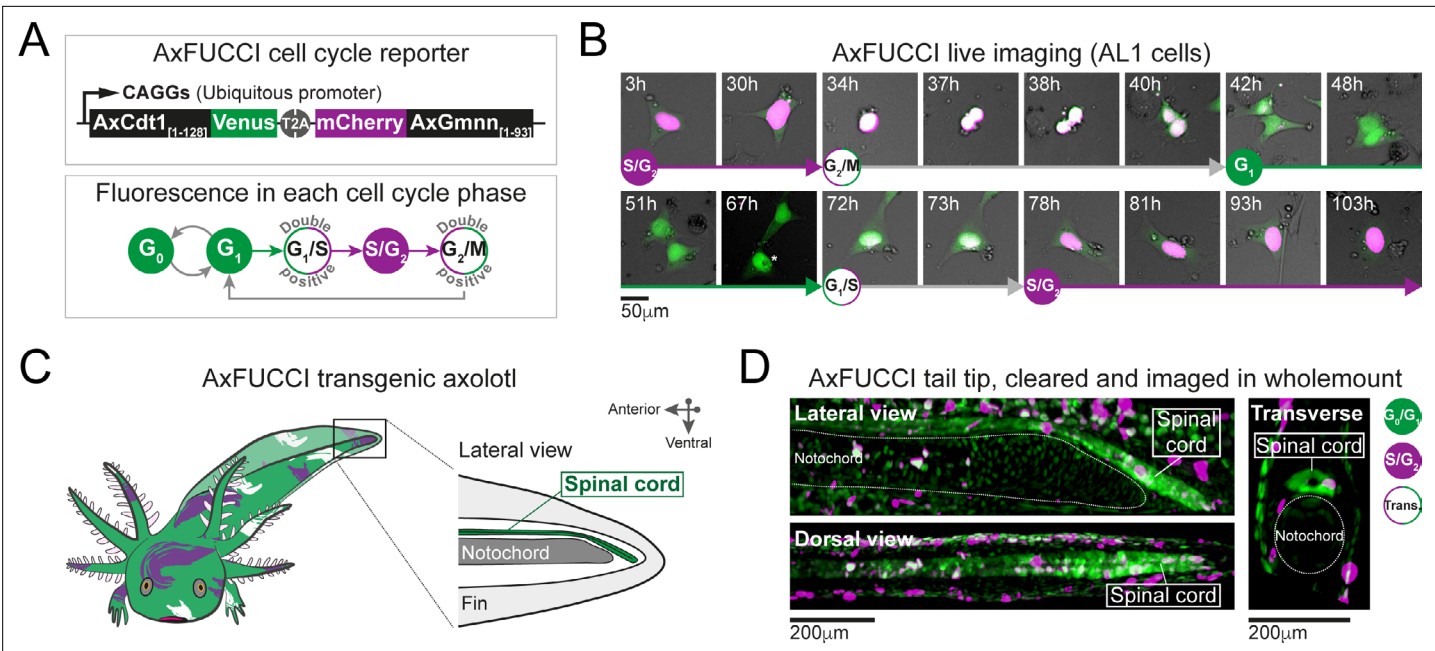

**Figure 3.** AxFUCCI – a transgenic cell cycle reporter for axolotl. (**A**) Top panel: AxFUCCI design. A ubiquitous promoter (CAGGs) drives expression of two AxFUCCI probes (AxCdt1[1-128]-Venus and mCherry-AxGmnn[1-93]) in one transcript. The two AxFUCCI probes are separated co-translationally by virtue of the viral 'self-cleaving' T2A peptide sequence. Bottom panel: AxFUCCI fluorescence combinations in each phase of the cell cycle. (**B**) Live imaging of a single AxFUCCI-electroporated axolotl cell in vitro. Each panel is a single frame acquired at the indicated hour (h) after the start of the imaging session. One cell cycle (from S/G$_2$ to the subsequent S/G$_2$) is depicted. At 67 hours, the two mitotic daughter cells moved apart; the asterisked daughter cell is depicted in the remaining panels. (**C**) Establishment of AxFUCCI transgenic axolotls. The location of the spinal cord is indicated in the context of the tail. (**D**) A fixed AxFUCCI tail, cleared and imaged in wholemount using lightsheet microscopy. The 3D data enable *post-hoc* digital sectioning of the same spinal cord into lateral, dorsal, or transverse views. Images depict maximum intensity projections through 50 μm (lateral and transverse views) or 150 μm (dorsal view) of tissue. Trans.: Transition-AxFUCCI (G$_1$/S or G$_2$/M transition).

The online version of this article includes the following figure supplement(s) for figure 3:

**Figure supplement 1.** Determination of Cdt1 and Gmnn fragments for AxFUCCI reporter.

**Figure supplement 2.** Fluorescence intensity measurements of AxFUCCI through the cell cycle.

**Figure supplement 3.** DNA quantification by flow cytometry to validate AxFUCCI functionality.

**Figure supplement 4.** Validation of AxFUCCI functionality through in vivo tissue analysis.

**Figure supplement 5.** The tissue clearing protocol does not alter spinal cord length.

analyzing FUCCI does not require cell dissociation (thus preserving spatial context within the tissue), immunostaining, or measurement of DNA content.

FUCCI has been adapted successfully to several model organisms, including mouse, zebrafish, *Drosophila,* and human cells (reviewed by *Zielke and Edgar, 2015*). We designed axolotl FUCCI de novo by extracting the degron-harboring sequences from the axolotl Cdt1 and axolotl Gmnn proteins. This was important as we found that the N-terminus of Cdt1 protein, harboring the PIP degron, is divergent across animal models (*Figure 3—figure supplement 1A*). We defined the relevant fragments of axolotl Cdt1 protein (harboring the PIP degron and Cy motif) and axolotl Gmnn protein (harboring the D box degron) using homology alignment and comparison with zebrafish FUCCI (*Figure 3—figure supplement 1A–D*; *Sugiyama et al., 2009*; *Bouldin et al., 2014*). The axolotl Cdt1$_{[aa1-128]}$ fragment was fused to mVenus fluorescent protein and the axolotl Gmnn$_{[aa1-93]}$ fragment to mCherry fluorescent protein. We used the CAGGs ubiquitous promoter and viral T2A sequence to co-express the Cdt1$_{[aa1-128]}$-mVenus and Gmnn$_{[aa1-93]}$-mCherry fusions in one transcript. The resulting axolotl-specific cell cycle reporter is referred to as AxFUCCI (*Figure 3A*).

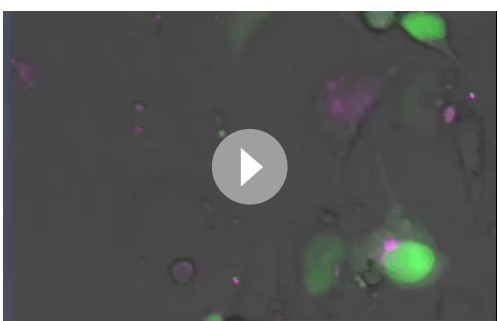

**Video 2.** Live imaging of an AxFUCCI-electroporated AL1 cell ($G_1$ to $G_1$). A single AxFUCCI-electroporated AL1 cell passing through one cell cycle from $G_1$ to $G_1$, imaged hourly over ~130h. The cell transitions through the cell cycle phases in the following order: Green ($G_0$/$G_1$-AxFUCCI) > White (Transition-AxFUCCI) > Magenta (S/$G_2$-AxFUCCI) > White (Transition-AxFUCCI) > Mitosis > Two Green daughter cells ($G_0$/$G_1$-AxFUCCI). This cell corresponds to Cell 41 tracked in *Figure 3—figure supplement 2*. Fluorescence intensity drops after mitosis due to fluorophore and plasmid dilution (non-integrating transgene).

https://elifesciences.org/articles/55665/figures#video2

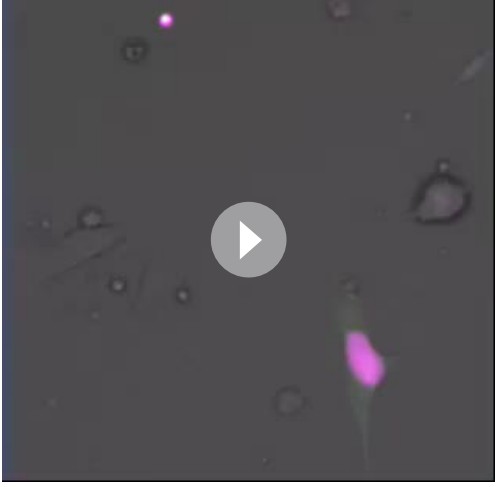

**Video 3.** Live imaging of an AxFUCCI-electroporated AL1 cell (S/$G_2$ to S/$G_2$). A single AxFUCCI-electroporated AL1 cell passing through one cell cycle from S/$G_2$ to S/$G_2$, imaged overly over ~120h. The cell transitions through the cell cycle phases in the following order: Magenta (S/$G_2$-AxFUCCI) > White (Transition-AxFUCCI) > Mitosis > Two Green daughter cells ($G_0$/$G_1$-AxFUCCI) > White (Transition-AxFUCCI) > Magenta (S/$G_2$-AxFUCCI). This cell corresponds to Cell 30 tracked in *Figure 3—figure supplement 2*. Fluorescence intensity drops after mitosis due to fluorophore and plasmid dilution (non-integrating transgene).

https://elifesciences.org/articles/55665/figures#video3

## AxFUCCI discriminates cell cycle phases faithfully in vivo

We performed live-cell imaging, DNA content analysis, and immunofluorescence-based characterizations to validate the ability of AxFUCCI to report the phases of the cell cycle. First, we electroporated an immortalized axolotl cell culture line (AL1 cells) with AxFUCCI plasmid and performed live imaging. As expected, AxFUCCI fluorescence oscillated during the cell cycle in the order mVenus > Double positive > mCherry > Double positive > mVenus (*Figure 3B*, *Figure 3—figure supplement 2A,B*, *Video 2* and *Video 3*). We confirmed that mVenus-positive cells were $G_0$/$G_1$ cells and that mCherry-positive cells were S/$G_2$ cells using a flow cytometer to analyze DNA content (*Figure 3—figure supplement 3A*). Cells are transiently AxFUCCI double positive between $G_0$/$G_1$ and S/$G_2$ (*Figure 3B* and *Figure 3—figure supplement 2A,B*); we infer that these cells are at the $G_1$/S boundary, as observed in mouse FUCCI (*Abe et al., 2013*). Interestingly, and in contrast to FUCCI in other model organisms, we observe a second AxFUCCI double-positive window between S/$G_2$ and $G_0$/$G_1$, corresponding to the $G_2$/M boundary and M cells (*Figure 3B*, *Figure 3—figure supplement 2A,B*). Thus, AxFUCCI discriminates the following cell cycle phases in axolotl: $G_0$/$G_1$ (mVenus only); S/$G_2$ (mCherry only); $G_1$/S transition and $G_2$/M transition (double positive) (*Figure 3B*). Importantly, AxFUCCI's capacity to label defined landmarks in the cell cycle (i.e., $G_1$/S and $G_2$/M transition) enabled us to later test for cell cycle synchronization, a characteristic feature of our model.

We generated stable transgenic AxFUCCI axolotls using I-SceI-mediated transgenesis, bred them to sexual maturity, and used $F_1$ (germline-transmitted) progeny for further validations (*Figure 3C*). AxFUCCI animals developed at a similar rate to their non-transgenic siblings, did not differ in their basal cell proliferation (*Figure 3—figure supplement 4A*), and regenerated amputated tail tissue with similar kinetics to the *d/d* animals used in our previous study (*Figure 3—figure supplement 4B*). Cells dissociated from AxFUCCI axolotl tails and analyzed using a flow cytometer exhibited the expected fluorescence/DNA content relationships (*Figure 3—figure supplement 4C*). As a second assay, we prepared spinal cord tissue sections from AxFUCCI axolotls and compared DNA content (as assessed by DAPI fluorescence) in mVenus versus mCherry-positive cells. As expected, mCherry-positive cells harbored significantly more DNA than mVenus-positive cells (*Figure 3—figure supplement 4D,E*).

Thirdly, we injected AxFUCCI axolotls intraperitoneally with EdU, a thymidine analogue that is incorporated into DNA during S phase. Following an 8 hours EdU pulse, mCherry-positive cells but not mVenus-positive cells should be EdU-positive and this was indeed the case (*Figure 3—figure supplement 4F*). Finally, we co-stained AxFUCCI spinal cord tissue sections with cell type-specific markers. NeuN-expressing neurons, located on the periphery of the spinal cord, are post-mitotic, differentiated cells ($G_0$) and should be mVenus-positive (and never mCherry-positive). Indeed, we found that 100% of neurons that expressed AxFUCCI were mVenus-positive (*Figure 3—figure supplement 4G*). By contrast, Sox2-expressing ependymal cells, which are proliferative cells, expressed mVenus, mCherry, or both fluorophores (*Figure 3—figure supplement 4H*).

Based on these validations, and for simplicity, we refer to mVenus fluorescence as '$G_0$/$G_1$-AxFUCCI' (green in *Figures 3–6*), mCherry fluorescence as 'S/$G_2$-AxFUCCI' (magenta in *Figures 3–6*), and double positivity as 'Transition-AxFUCCI' (white in *Figures 3–6*).

## Measuring the recruitment zone in vivo using AxFUCCI

We used AxFUCCI animals to measure the size of the ependymal cell recruitment zone in vivo. We amputated AxFUCCI tails at 5 mm from the tail tips and harvested replicate regenerating tails daily up to 5 days post-amputation. We implemented a pipeline to optically clear and image fixed AxFUCCI tail tissue in wholemount, which enabled *post-hoc* digital re-sectioning of the imaging data into any orientation for accurate measurement (*Figure 3D*; *Pende et al., 2020*; *Video 4*). Importantly, this pipeline did not alter the length of the tail tissue (*Figure 3—figure supplement 5*). We found that, at amputation, most ependymal cells expressed $G_0$/$G_1$-AxFUCCI and only a minority expressed S/$G_2$-AxFUCCI (*Figure 4A*). In the 5 days following tail tip amputation, the proportion of S/$G_2$-AxFUCCI-expressing cells increased locally and sharply at the amputation site, then propagated anteriorly along the spinal cord, consistent with the appearance of a recruitment zone (*Figures 1B and 4A*).

As a first step, we manually quantified the percentage of ependymal cells expressing either $G_0$/$G_1$-AxFUCCI alone or S/$G_2$-AxFUCCI alone within the 1000–1600 µm of spinal cord anterior to the injury site at each day post-amputation. A 1600 µm length should encompass not only the recruitment zone but also more anteriorly located ependymal cells that are not recruited by the injury signal and that continue to cycle slowly (*Figure 1B*). We performed our quantifications in 100 µm adjacent bins to preserve spatial information using the severed notochord tip to denote the amputation plane. To test statistically whether the cells expressing $G_0$/$G_1$-AxFUCCI and S/$G_2$-AxFUCCI are heterogeneously distributed along the AP axis in the regenerating spinal cords (i.e., if a recruitment zone can be detected), we followed an approach similar to that that we used previously to determine the switchpoint (*Rost et al., 2016*). We fitted the measured spatial AP profiles of the percentage of $G_0$/$G_1$-AxFUCCI and S/$G_2$-AxFUCCI-expressing cells in each animal with a mathematical model assuming two adjacent homogeneous spatial zones separated by an AP border, which we assumed was the same for both $G_0$/$G_1$ and S/$G_2$-AxFUCCI data. For each animal at each timepoint, we tested if the mean percentage of cells expressing $G_0$/$G_1$-AxFUCCI or S/$G_2$-AxFUCCI in the anterior versus the posterior zones was significantly different by running a Kolmogorov–Smirnov test. Up until 3 days post-amputation, no statistical significance was detected between anterior and posterior in the $G_0$/$G_1$-AxFUCCI and S/$G_2$-AxFUCCI data (*Figure 4B*, *Figure 4—figure supplement 1A,B,C,D* and *Figure 4—figure supplement 2A,B,C,D*). By contrast, at 4 and 5 days post-amputation, $G_0$/$G_1$-AxFUCCI and S/$G_2$-AxFUCCI data revealed a significant difference between the anterior and posterior zones, consistent with the appearance of a recruitment zone (*Figure 4B*, *Figure 4—figure supplement 1E,F* and *Figure 4—figure supplement 2E,F*).

Crucially, we measured the AP border of the AxFUCCI data to be at –717 ± 272 µm relative to the amputation plane on day 4 and –446 ± 112 µm on day 5, overlapping within 2 sigma the –782 ± 50 µm and –710 ± 62 µm recruitment limits predicted by our model (*Figure 4B*). Moreover, the appearance of the recruitment zone between days 3 and 4 post-amputation accommodates the 85 hours recruitment time in our model. Thus, AxFUCCI animals confirmed the predicted appearance time and size of the recruitment zone.

## Regenerating cells have high cell cycle synchrony in vivo

Our model of $G_1$ shortening (*Figure 1C*, *Figure 1—figure supplement 2*) predicts that ependymal cells in the recruitment zone should exhibit high synchrony with each other in the cell cycle during

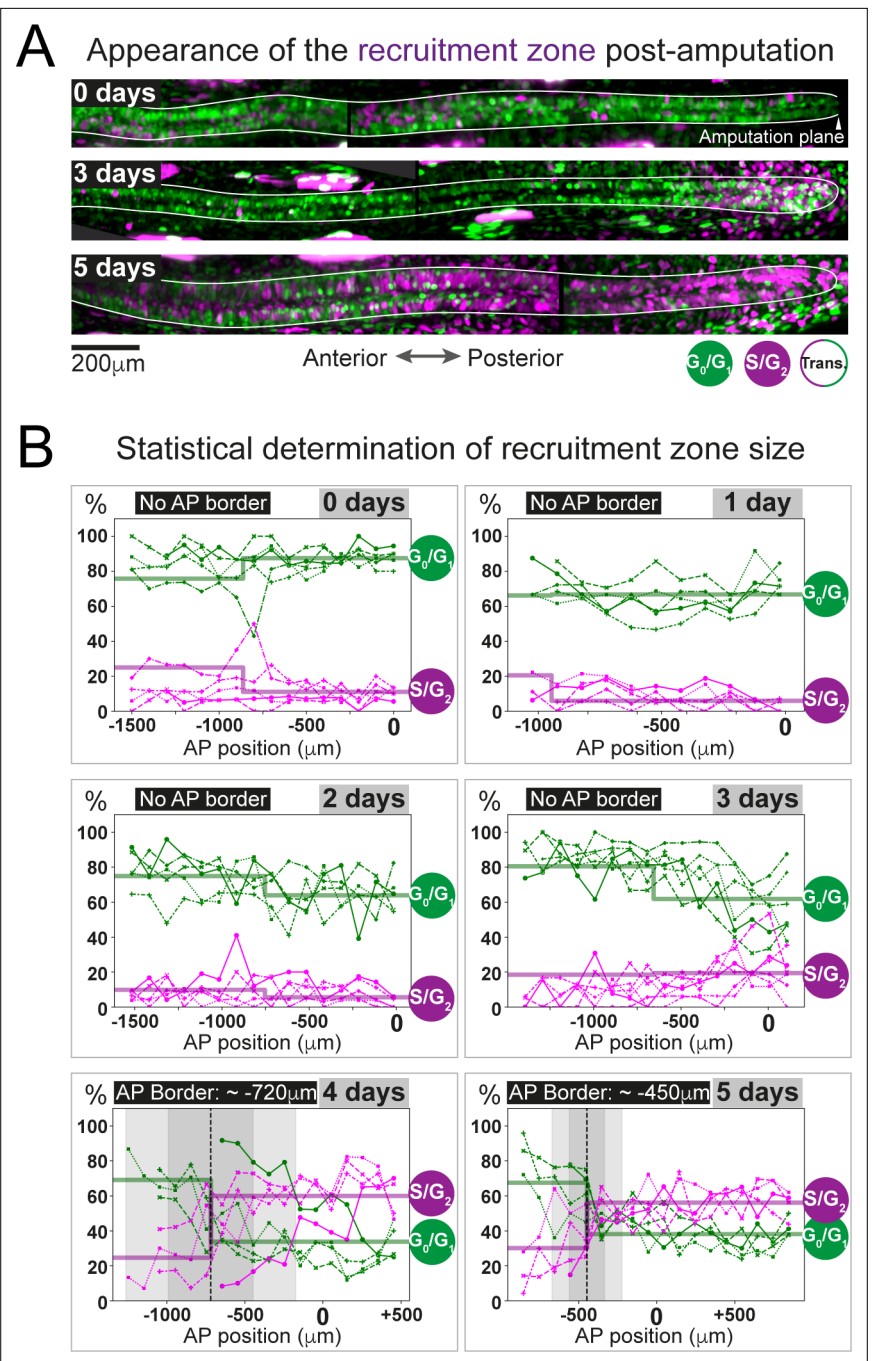

**Figure 4.** The predicted recruitment zone size is observed in AxFUCCI tails after amputation. (**A**) Visualization of the recruitment zone in AxFUCCI tails. At amputation, most ependymal cells are $G_0/G_1$-AxFUCCI positive (green). After amputation, a decrease in $G_0/G_1$-AxFUCCI and an increase in $S/G_2$-AxFUCCI (magenta) expression is seen anterior to the amputation plane, and this zone increases in size anteriorly through day 5 post-amputation. The spinal cord is outlined. Images depict maximum intensity projections through 30 μm of tissue and are composites of two adjacent fields of view. (**B**) Quantification of the anterior-posterior border (AP border) that delimits the recruitment zone in $G_0/G_1$ and $S/G_2$-AxFUCCI data. Percentage of $G_0/G_1$ (in green) and $S/G_2$ (in magenta)-AxFUCCI-expressing cells quantified in 100 μm bins along the AP spinal cord axis. A mathematical model assuming two adjacent spatially homogeneous zones separated by an AP border was fitted to the $G_0/G_1$ and $S/G_2$-AxFUCCI-data for each animal. Significant differences between anterior and posterior zones were detected only at days 4 and 5 (Kolmogorov–Smirnov test p=0.0286). The AP border mean and 2 sigmas are depicted as a black dashed line and a gray shaded areas, respectively. AP position is defined with respect to the amputation plane (0

*Figure 4 continued on next page*

*Figure 4 continued*

µm). n = 4–6 tails per time point, ~300 cells each. Different symbols depict different animals; each line represents one animal. Best-fitting values of the model regarding the anterior and posterior percentage of AxFUCCI data are in *Figure 4—figure supplement 1*. Individual fittings are in *Figure 4—figure supplement 2*. For more details, see Section 2.12.

The online version of this article includes the following figure supplement(s) for figure 4:

**Figure supplement 1.** Estimation of anterior and posterior (AP) percentages of $G_0/G_1$-AxFUCCI and $S/G_2$-AxFUCCI-expressing cells by fitting a two-zones model.

**Figure supplement 2.** Individual fitting of the two-zones model to experimental AP profiles of the percentages of $G_0/G_1$-AxFUCCI and $S/G_2$-AxFUCCI-expressing cells.

early regeneration, a property that has not been investigated. AxFUCCI axolotls enabled us to assess for potential ependymal cell cycle synchrony in regenerating spinal cord.

We performed a more rigorous quantification of cell cycle distribution from our wholemount data in which we focused on the 600 µm of spinal cord immediately anterior to the amputation plane, within the recruitment zone, and additionally including Transition-AxFUCCI ($G_1/S$ and $G_2/M$ transition cells) and M phase cells. At the moment of amputation, 85% ± 5% of ependymal cells expressed $G_0/G_1$-AxFUCCI and 11% ± 6% expressed $S/G_2$-AxFUCCI (*Figure 5A*). We note that this baseline differs from the one that we reported previously in smaller, *d/d* control animals, but we found this lower basal proliferation rate to be consistent among animals in this study independent of genotype (*Figure 3—figure supplement 4A*; *Rodrigo Albors et al., 2015*). Restrictions in feeding and/or changes in animal handling during the COVID-19 pandemic might explain this difference (see Discussion). As we expected our model might be robust to the baseline cell cycle profile, we gathered measurements in these animals for further analysis.

After amputation, we detected a significant drop in $G_0/G_1$-AxFUCCI-expressing ependymal cells already at day 1. Reciprocally, the number of $S/G_2$-AxFUCCI-expressing cells increased significantly starting at 3 days and reached 50% at days 4 and 5 post-amputation, compared to a baseline percentage below 10% (*Figure 5A and B*). M phase cells started to appear noticeably from day 4, although it was not possible to perform statistical analysis due to the absence of M phase cells in the 0 day post-amputation samples (*Figure 5B*). Intriguingly, we observed a transient 'burst' of Transition-AxFUCCI cells at days 1 and 2 post-amputation. The percentage of Transition-AxFUCCI cells increased to 21% ± 12% and 23% ± 12%, respectively, during this burst, before declining back to the baseline level of 5% ± 4% at 3 days (*Figure 5B*). We confirmed the accuracy of these quantifications by preparing tissue sections from replicate AxFUCCI spinal cords and imaging them using confocal microscopy (*Figure 5C*, *Figure 5—figure supplement 1A*).

Transition-AxFUCCI could correspond to $G_1/S$ transition or $G_2/M$ transition (*Figure 3A*). We hypothesized that Transition-AxFUCCI cells at 1 and 2 days post-amputation resided at the $G_1/S$ transition as they appeared between the decline in $G_0/G_1$-AxFUCCI cells at day 1 and the increase in $S/G_2$-AxFUCCI cells at day 3. To confirm this, we subjected AxFUCCI animals to an EdU pulse for 24 hours immediately prior to tail harvesting at 0, 1, or 2 days post-amputation. In this assay, Transition-AxFUCCI cells that incorporate EdU should be in $G_2/M$, while those that do not incorporate EdU should be in $G_1/S$ (*Figure 5—figure supplement 1B*). Consistent with our expectations, Transition-AxFUCCI cells at 1 and 2 days post-amputation were almost entirely EdU-negative and therefore resided in $G_1/S$ (*Figure 5D*, *Figure 5—figure supplement 1C,D*).

In sum, AxFUCCI revealed the following cell cycle dynamics during regeneration (summarized in *Figure 5E*). Most ependymal cells in the uninjured spinal cord reside in $G_0/G_1$ phase of the cell cycle. Following tail amputation, ependymal cells start to leave $G_0/G_1$ within the first day of amputation, transit through $G_1/S$ at days 1 and 2, enter $S/G_2$ from day 3 onwards, and undergo mitosis from day 4. The fact that these behaviors are readily observed at the population level indicates a high level of synchrony among ependymal cells in the recruitment zone. The $G_1/S$ transition acts as a discrete landmark in the cell cycle at which this synchrony can be inferred reliably. Transition-AxFUCCI cells are very rare (~5%) at amputation. We take the 4.5-fold increase in Transition-AxFUCCI-expressing cells at days 1 and 2 post-amputation as a strong indication of cell cycle synchrony during early spinal cord regeneration, a key prediction of our model.

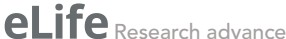

**Figure 5.** Ependymal cells exhibit cell cycle synchrony during spinal cord regeneration. (**A**) Cell cycle distributions of ependymal cells during the first 5 days of regeneration. Maximum intensity projections through 25 μm of spinal cord oriented anterior to left and posterior to right. The spinal cord is outlined. The lumen is in the center. Images were taken from within the 600 μm of most posterior regenerating spinal cord. (**B**) AxFUCCI data separated by cell cycle phase. Percentage of ependymal cells in each cell cycle phase in the 600 μm of most posterior regenerating spinal cord between 0 and 5 days post-amputation. Cells in mitosis (M) were counted independently of AxFUCCI and were identified based on their condensed chromatin as revealed by staining with DAPI (not shown). The percentage of AxFUCCI-negative cells ('-') was negligible and did not change at any time point (p=0.40, Wilcoxon rank sum). Each dot represents data from one tail. n = 4–6 tails per time point, ~100 cells each. Error bars indicate standard deviations. Day 0 data (immediately after amputation) were taken as baselines, and statistical analyses were performed against these baselines. Kruskal–Wallis tests followed by Wilcoxon rank sum tests revealed a significant decrease in $G_0/G_1$-AxFUCCI from 1 day post-amputation (p=0.02) and a significant increase in $S/G_2$-AxFUCCI starting at 3 days post-amputation (p=0.02). One-way ANOVA followed by Tukey's HSD test revealed a significant increase in Transition-AxFUCCI cells at 1 and 2 days post-amputation (p=0.04 and 0.02, respectively), but not at later days post-amputation (p>0.99). Mitotic cells were absent in the 0 day samples, precluding statistical analysis. ns: not statistically significant. (**C**) The emergence of Transition-AxFUCCI cells at 1 and 2 days post-amputation was confirmed in tissue sections. Images are single-plane confocal images of spinal cord cross-sections fixed at the indicated times after amputation. Percentages indicate the percentage of Transition-AxFUCCI cells at each time point (mean ± standard deviation). n = 3 tails per time point, ~400 cells counted in each, corresponding to ~750 μm of spinal cord. (**D**) Transition-AxFUCCI-expressing cells at 1 and 2 days post-amputation reside at the $G_1/S$ transition. Transition-AxFUCCI cells can be either at the $G_1/S$ transition or the $G_2/M$ transition. Following an 8 hours (0 days) or 24 hours (1 and 2 days) EdU pulse, only $G_2/M$ cells should become labeled with EdU. Tails were fixed, then processed for tissue sectioning and EdU detection. >97% of Transition-AxFUCCI cells were EdU-negative at 1 and 2 days post-amputation, indicating that they reside at the $G_1/S$ transition. n = 3 tails per time point. A total of 36, 442, and 315 Transition-AxFUCCI cells were assayed at 0, 1, and 2 days post-amputation, respectively. (**E**) Cell cycle dynamics during axolotl spinal cord regeneration.

The online version of this article includes the following figure supplement(s) for figure 5:

**Figure supplement 1.** Transition-AxFUCCI cells at 1 and 2 days post-amputation reside at the $G_1/S$ boundary.

## A convergence in regenerative response from distinct baselines

The AxFUCCI quantifications validated key predictions of our model in terms of the size of the ependymal cell recruitment zone and in demonstrating high cell cycle synchrony during the regenerative response. We were intrigued that these agreements occurred despite a significant difference

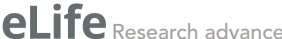

**Figure 6.** Convergence of cell cycle response between the model simulations and the AxFUCCI data. Heatmaps depicting spatiotemporal distribution of $G_0/G_1$-AxFUCCI cells (**A**), $S/G_2$-AxFUCCI cells (**B**), model-predicted cells in $G_1$ phase (**C**), and model-predicted cells in S phase (**D**). x-axes depict anterior-posterior (AP) position with respect to the amputation plane (AP position = 0 μm). y-axes depict time (days) post-amputation and both experimental and simulated replicates within each time point. The color codes correspond to the percentage of cells in the corresponding phase. Transition-AxFUCCI data are in *Figure 6—figure supplement 2*. (**E**) Model-predicted occurrence of $G_1$-S transitions is more often within the recruited cells (gray circles) compared to the non-recruited (orange crosses) cells. At days 4 and 5, the recruitment limits overlap with the AP borders. Independent simulations from 10 independent seeds are shown and the model is parameterized as in *Figure 2*.

The online version of this article includes the following figure supplement(s) for figure 6:

**Figure supplement 1.** Modeled spatiotemporal distributions of ependymal cell divisions during axolotl spinal cord regeneration.

**Figure supplement 2.** Spatiotemporal distributions of Transition-AxFUCCI cells during axolotl spinal cord regeneration.

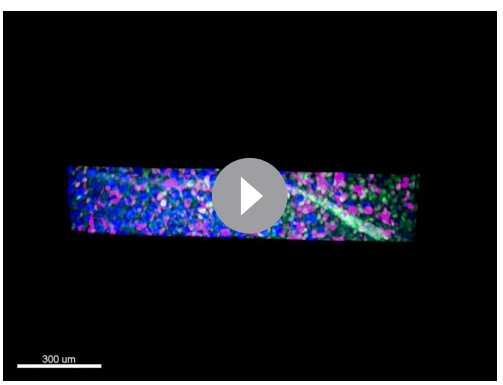

**Video 4.** 3D imaging of an optically cleared AxFUCCI tail tip. Volume rendering of a fixed and optically cleared AxFUCCI tail tip, co-stained for DNA using DAPI (blue) and imaged with a lightsheet microscope. The spinal cord is visible as an intensely green and magenta rod at the center of the sample. Peripheral signal is AxFUCCI expression in surface epidermal cells. Other internal structures, including notochord, have weaker AxFUCCI expression and are not visible in this rendering. Volume rendering was performed using Imaris software. See also *Figure 3D*.

https://elifesciences.org/articles/55665/figures#video4

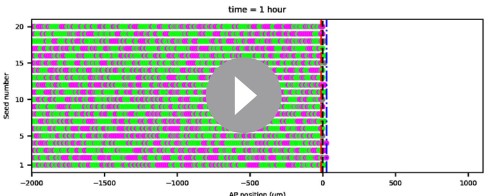

**Video 5.** 1D Model simulations of spinal cord regeneration showing cells in $G_1$ and S phases. 20 model simulations from 20 different random seeds (the seeds are shown in the vertical axis). Cells in $G_1$ and S are depicted in green and magenta, respectively. Predicted recruitment limit $\xi(t)$ and outgrowth are indicated with the vertical red and blue discontinuous lines, respectively. Amputation plane is depicted as a vertical black dashed line. The simulations have the same parametrization than the simulations showed in *Figure 2A and B*.

https://elifesciences.org/articles/55665/figures#video5

in starting cell cycle conditions (day 0) between the AxFUCCI animals and our model, which was based on data described in *Rodrigo Albors et al., 2015*. The AxFUCCI animals were of size 5.5 cm from snout to tail and, on the day of amputation, 11% ± 6% of ependymal cells expressed S/$G_2$-AxFUCCI. In contrast, our model was parametrized using measurements acquired from 3 cm, non-transgenic axolotls, in which the baseline percentage of S-phase cells was four-fold higher, as inferred from cumulative BrdU labeling (*Rodrigo Albors et al., 2015*). Despite these differences, we found the speed of spinal cord tissue regeneration to be similar in the two datasets (*Figure 3—figure supplement 4B*). This could reflect a convergence in regenerative response at the cell cycle level.

We investigated further the kinetics of this convergence by using our model to generate day-by-day simulations of the spatial distribution of cells in $G_1$ or S phase and comparing these simulations to the cell cycle phase quantifications made from the AxFUCCI animals (*Figure 6A–D*, *Video 5*). We validated the simulations by testing where and when ependymal cells would leave $G_1$ phase to enter S phase and found that, as expected, simulated $G_1$-to-S transitions were more frequent posterior to the experimentally derived recruitment limit (*Figure 6E*), which at days 4 and 5 overlap with the determined AP borders. Consequently, simulated cell divisions (M phase) exhibited a similar pattern (*Figure 6—figure supplement 1*).

We compared the model simulations to the AxFUCCI data. As noted, the cell cycle conditions at day 0 differ between the AxFUCCI experiments and our model but match quantitatively at day 4 post-amputation (*Figure 6A–D*, *Video 5*). We found the appearance of the recruitment zone to be later in the AxFUCCI animals (days 4–5) than in the simulations (day 2). However, once the recruitment zone is evident in the AxFUCCI data, its size at days 4 and 5 is comparable to the one predicted by the simulations at those times. Thus, the recruitment zone in the AxFUCCI spinal cord manifests more synchronously and rapidly than in the simulations, in which it increases gradually in size in a posterior-to-anterior direction between days 2 and 4. This is likely a consequence of the larger pool of $G_0/G_1$ cells in the AxFUCCI animals at days 0–2 compared to the model, which incurs a collective (but relatively synchronous) lag in S phase entry (recruitment zone manifestation). Our data reveal two contrasting trajectories towards achieving a common regenerative output.

## Discussion

The tissue response to spinal cord injury differs greatly across vertebrates. In mammals, including humans, injuries to the spinal cord often result in permanent tissue damage. In salamanders like the axolotl, however, the ependymal cell response is tightly orchestrated to faithfully rebuild the missing spinal cord (*Joven and Simon, 2018*; *Tazaki et al., 2017*). Following tail amputation, ependymal cells in the axolotl spinal cord switch from slow, neurogenic to faster, proliferative cell divisions (*Rodrigo Albors et al., 2015*). These faster cell cycles lead to the expansion of the ependymal/neural stem cell pool and drive an explosive regenerative outgrowth. However, the mechanisms regulating cell cycle dynamics during regeneration are not fully understood. Here, by using a modeling approach tightly linked to experimental data, we find that the spatiotemporal pattern of cell proliferation in the regenerating axolotl spinal cord is consistent with a signal that propagates anteriorly 828 µm from the injury site during the first 85 hours post-amputation. Although, for simplicity, we refer in this article to a single amputation-induced signal, our model could naturally extend to the combined output of multiple molecular and/or biophysical signals. We show that shortening of S phase is sufficient to explain the explosive growth observed during the first days of regeneration, but that both S and $G_1$ shortening are necessary to explain/sustain further outgrowth before the first newborn neurons are seen (*Rodrigo Albors et al., 2015*).

Compared to the number of mathematical models designed to unveil pattern formation phenomena during development (*Morelli et al., 2012*), modeling of regeneration is still in its infancy (*Chara et al., 2014*). An interesting example of modeling applied to regenerative processes was given by a system of deterministic ordinary differential equations that was superbly used to disentangle how secreted signaling factors could be used to control the output of multistage cell lineages in a self-renewing neural tissue, the mammalian olfactory epithelium (*Lander et al., 2009*). Another mathematical model based on ordinary differential equations was conceived to establish the causal relationship between the individually quantified cellular processes to unravel the stem cell dynamics in the developing spinal cord in chick and mouse (*Kicheva et al., 2014*). In a similar approach, we previously modeled the regenerating axolotl spinal cord by means of a system of deterministic ordinary differential equations describing the kinetics of the cycling and quiescent ependymal cell numbers, which we mapped to a model of spinal cord outgrowth (*Rost et al., 2016*). This allowed us to conclude that while cell influx and cell cycle re-entry play a minor role, the acceleration of the cell cycle is the major driver of regenerative spinal cord outgrowth in axolotls (*Rost et al., 2016*). A more recent study based on ordinary and partial differential equations involving cell proliferation was used to predict the spinal cord growth of the knifefish (*Ilieş et al., 2018*).

In this study, we investigated the spatiotemporal distribution of cell proliferation during axolotl spinal cord regeneration. To do so, and in contrast with the aforementioned articles, we developed a more general and yet accurate cell-based model introducing the spatial dimension relevant to the problem: the AP axis. To further build a more realistic model, we included non-deterministic attributes: an exponential distribution of the initial coordinates along the cell cycle and a lognormal distribution of the cell cycle length. In the model, a signal shortens the cell cycle of ependymal cells along the AP axis by shortening their $G_1$ and S phases, as we reported earlier (*Rodrigo Albors et al., 2015*). Regulation of $G_1$ and S phases are well-known mechanisms controlling cell fate and cell output in a number of developmental contexts. In the brain, $G_1$ lengthening results in longer cell cycles in neural progenitors undergoing neurogenesis (*Lukaszewicz et al., 2005*; *Calegari et al., 2005*; *Takahashi et al., 1995*), while experimental shortening of $G_1$ in neural progenitors of the cerebral cortex results in more proliferative divisions, increasing the progenitor pool and delaying neurogenesis (*Salomoni and Calegari, 2010*; *Lange et al., 2009*; *Pilaz et al., 2009*; *Calegari and Huttner, 2003*). Here, we have shown that the shortening of $G_1$ during spinal cord regeneration is necessary to sustain the expansion of the ependymal cell pool. Together, these findings point to the regulation of $G_1$ length as a key mechanism regulating the output of neural stem/progenitor cell divisions in development and in regeneration. The length of S phase is also regulated during development by modulating the number of DNA replication origins (*Nordman and Orr-Weaver, 2012*). In mammals, shortening of S phase seems to play a role in regulating the mode of cell division: mouse neural progenitors committed to neurogenesis and neurogenic cortical progenitors in the ferret undergo shorter S phase than their self-renewing/proliferative counterparts (*Turrero García et al., 2016*; *Arai et al., 2011*). In the axolotl, regenerating ependymal cells shorten S phase during the expansion/outgrowth phase.

Together, these findings suggest that the regulation of S phase controls cell output in the context of development and regeneration rather than influence the mode of cell division. The combined shortening of S and $G_1$ in the regenerating spinal cord sustains the expansion of the resident ependymal/neural stem cell pool at the expense of neurogenesis. In this line, experimentally shortening $G_1$ and S phases in cortical progenitors of the developing mouse brain delayed the onset of neurogenesis (*Hasenpusch-Theil et al., 2018*). Our findings add to the evidence that cell cycle regulation is a key mechanism controlling the number and type of cells needed to generate and regenerate a tissue.

Another prediction of our model is that a signal must spread about 800 µm from the injury site while recruiting ependymal cells during the 85 hours after amputation to explain the spatiotemporal pattern of cell proliferation in the regenerating spinal cord. In order to test this prediction experimentally, we adapted FUCCI technology to axolotls, which enabled us to visualize cell cycle dynamics in vivo. We found remarkable agreement between our prediction and the size and timing of appearance of the recruitment zone in AxFUCCI spinal cords. Our prediction was made based on data from 3 cm snout-to-tail axolotls, while the AxFUCCI measurements were taken from 5.5 cm axolotls. That the size of the recruitment zone appears similar between these two animal sizes could be important in understanding the identity of the injury-induced signal and how it spreads to recruit ependymal cells. Future experiments will determine if the size of the recruitment zone also remains constant in even larger axolotls.

A characteristic feature of our model is that $G_1$ shortening after amputation causes ependymal cells to partially synchronize with one another as they pass through $G_1$. Cell cycle synchronization is difficult to measure in cells in vivo. Here, AxFUCCI's property of labeling short, discrete landmarks in the cell cycle (e.g., $G_1$/S transition) enabled us to visualize high $G_1$/S synchrony at 1 and 2 days post-amputation in vivo. It will be interesting to assess whether a similar phenomenon occurs during regeneration of other tissues in the axolotl (e.g., limb) and in other regenerative organisms.

Although we observed an excellent match between our model simulations and the AxFUCCI data in terms of recruitment zone size at days 4 and 5 post-amputation, we also encountered quantitative differences at earlier time points. In particular, we found that significantly fewer ependymal cells were in S/$G_2$ at baseline conditions in the present study compared to our previous study (*Rost et al., 2016*). The AxFUCCI experiments reported here were carried out under COVID-19-related operational restrictions – in particular, animal feeding frequency was reduced. A dietary reduction could plausibly reduce baseline ependymal cell proliferation rate and animal growth. We also cannot exclude an impact from general housing conditions as the previous experiments were performed in a different animal facility with, for example, a different water supply. We note, however, that the baseline of S/$G_2$ ependymal cells that we observe in this study is in good agreement with recent results obtained from FUCCI axolotls that were independently generated using zebrafish DNA constructs (*Duerr et al., 2021*). From the data analysis side, it is important to note that in our AxFUCCI experiments $G_0$/$G_1$ AxFUCCI cells become Transition-AxFUCCI cells and then S/$G_2$ AxFUCCI cells. This means that transition cells could be either cells in the late $G_1$ or in the early S phases. Modeled cells, in contrast, go straight from $G_1$ to S phase. Consequently, we cannot quantitatively equate the 'Transition phase' cells between experiments and model. This is why the AxFUCCI data and the model simulations can only be qualitatively compared, especially at days 1 and 2, when transition cells peak. Additionally, given that the model is parametrized using cell cycle phase lengths, which are not possible to infer from the AxFUCCI data (which measure cell cycle phase proportions), it was not trivial to re-parametrize and re-run our model using the new data generated in this study. Despite these considerations, we found that the proportions of ependymal cells in the $G_0$/$G_1$ versus S/$G_2$ AxFUCCI data at 4 and 5 days post-amputation – that is, during the first regenerative cell cycle – accurately and quantitatively matched the simulations of our model. Moreover, the outgrowth rate of the regenerating spinal cord was consistent between the AxFUCCI animals in this study and the animals measured in our previous study (*Figure 3—figure supplement 4B*). This is to say that axolotls mount a remarkably consistent regenerative response within their first cell cycle after amputation, possibly converging their cell cycle responses despite differences in baseline proliferation. It will be fascinating to investigate the molecular mechanisms that enable this consistent regenerative response in axolotls across age/size and nutrition availability.

An important question now is whether the spatiotemporal cell cycle response observed in this study agrees with known signaling events operating during spinal cord regeneration. A strong candidate

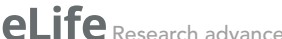

molecule for recruiting ependymal cells is the axolotl MARCKS-like protein (AxMLP), a secreted factor involved in the proliferative response during axolotl appendage regeneration (*Sugiura et al., 2016*). AxMLP is normally expressed in spinal cord cells but is upregulated following tail amputation, peaking 12–24 hours after amputation and returning to basal levels a day later (*Sugiura et al., 2016*). The timing prediction of our model is in agreement with the peak of AxMLP followed by a downstream period of signal decoding to instruct intrinsic cellular changes that lead to faster cell cycles. Moreover, the secreted nature of AxMLP protein could explain the long-range proliferative response in the regenerating spinal cord. In the future, a tighter time-course characterization of AxMLP localization throughout axolotl spinal cord regeneration will help to put our predictions to test.

Changes in the biophysical properties of the amputated tail could also trigger the orderly increase in cell proliferation. In *Xenopus* tadpoles, tail amputation leads to the activation of the $H^+$ V-AT-Pase, which is necessary and sufficient to promote tail regeneration (*Adams et al., 2007*). In the axolotl, tail amputation triggers changes in calcium, sodium, and membrane potential at the injury site (*Ozkucur et al., 2010*) while spinal cord transection induces a rapid and dynamic change in the resting membrane potential, which drives a c-Fos-dependent gene expression program promoting a pro-regenerative response (*Sabin et al., 2015*). The proliferation-inducing signal could also be of mechanical nature (*Chiou and Collins, 2018*). In this direction, it is interesting that spinal cord transection in the zebrafish induces an immediate alteration in the mechanical properties in the lesion site, which gradually returns to normal (*Schlüßler et al., 2018*). Our predictions of the temporal and spatial distribution that such proliferation-inducing signal could have will guide efforts to narrow down the mechanisms responsible for successful spinal cord regeneration.

Taken together, our study provides a finer mechanistic understanding of the cell cycle kinetics that drive spinal cord regeneration in axolotl and paves the way to search for the signal or signals that launch the successful ependymal cell response to spinal cord injury.

## Materials and methods

**Key resources table**

| Reagent type (species) or resource | Designation | Source or reference | Identifiers | Additional information |
|---|---|---|---|---|
| Strain, strain background (*Ambystoma mexicanum*, *d/d* strain) | Axolotl, *d/d* strain | Tanaka lab axolotl colony | - | Axolotl stock maintained in Tanaka lab, Vienna, Austria. |
| Genetic reagent (*Ambystoma mexicanum*, AxFUCCI transgenic) | Transgenic AxFUCCI axolotl | This paper | - | Transgenic stock generated in *d/d* genetic background in Tanaka lab, Vienna, Austria. |
| Gene (*Ambystoma mexicanum*) | Cdt1 | Axolotl genome release v6.0-DD *Schloissnig, 2021* | CDT1\|AMEX60DD201052583.1 | Axolotl genome release v6.0-DD available at: https://www.axolotl-omics.org/assemblies. |
| Gene (*Ambystoma mexicanum*) | Gmnn | Axolotl genome release v6.0-DD *Schloissnig, 2021* | GMNN\|AMEX60DD301038574.1 | Axolotl genome release v6.0-DD available at: https://www.axolotl-omics.org/assemblies. |

*Continued on next page*

*Continued*

| Reagent type (species) or resource | Designation | Source or reference | Identifiers | Additional information |
|---|---|---|---|---|
| Cell line (AL1, *Ambystoma mexicanum*) | AL1 cells | *Roy et al., 2000* | - | AL1 cells were kindly provided by Dr David Gardiner (UC Irvine). AL1 cells are an immortalized mesenchymal axolotl line originally derived from a limb blastema. |
| Recombinant DNA reagent | Addgene plasmid mVenus N1 | Addgene | RRID:Addgene_27793 | To amplify coding sequence for mVenus fluorescent protein by PCR. |
| Recombinant DNA reagent | pAxFUCCI | This paper | pAxFUCCI | Plasmid encoding AxFUCCI. Used to generate AxFUCCI transgenic axolotls by I-SceI-mediated transgenesis. |
| Sequence-based reagent | axCdt1-fwd | This paper | PCR primer | TCAT**GCGGCCGC**ATGGCCCAGCTCCGGATGA. Forward primer to amplify axCdt1[aa1-128] from axolotl embryonic cDNA. Bold indicates a NotI restriction site. |
| Sequence-based reagent | axCdt1-rev | This paper | PCR primer | TCAT**GCTAGC**<u>GAATTCTCCGCTTCCTGCTGCGCTTCCTGCGCTTCC</u>CAGGGATGATGGG GTTAATGGCT Reverse primer to amplify axCdt1[aa1-128] from axolotl embryonic cDNA. Bold indicates a NheI restriction site. Underlined sequence encodes GS-rich linker. |
| Sequence-based reagent | Venus-fwd | This paper | PCR primer | TCAT**GCTAGC**ATGGTGAGCAAGGGCGAGG Forward primer to amplify mVenus sequence from addgene plasmid #27793. Bold indicates a NheI restriction site. |
| Sequence-based reagent | Venus-rev | This paper | PCR primer | TCAT**ACCGGT**CTTGTACAGCTCGTCCATGCC Reverse primer to amplify mVenus sequence from addgene plasmid #27793. Bold indicates a AgeI restriction site. |
| Sequence-based reagent | axGmnn-fwd | This paper | PCR primer | TCAT**TCCGGA**<u>GGAGGAGGAGGAAGCGGAGGAGGAGGAAGC</u>ATGAATGCTAAGAAAG CAGCGACAAT Forward primer to amplify axGmnn[aa1-93] from axolotl embryonic cDNA. Bold indicates a BspEI restriction site. Underlined sequence encodes GS-rich linker. |
| Sequence-based reagent | axGmnn-rev | This paper | PCR primer | TCAT**ATCGAT**GCTGTAAGCTTCTTGGGACA CCC Reverse primer to amplify axGmnn[aa1-93] from axolotl embryonic cDNA. Bold indicates a ClaI restriction site. |
| Antibody | Mouse monoclonal antibody to NeuN | Millipore | MAB377 RRID:AB_2298772 | 1:500 |
| Antibody | Rabbit polyclonal antibody to Sox2 | *Fei et al., 2016* | - | 1:1000 |
| Commercial assay or kit | Click-iT 647 EdU detection kit for Imaging | Thermo Fisher Scientific | C10340 | To label cells that have transited through S-phase. |
| Chemical compound, drug | Easyindex | LifeCanvas Technologies | - | Refractive index matching solution for light sheet imaging (RI 1.46). |
| Software, algorithm | Trackmate plugin for Fiji | *Tinevez et al., 2017* | - | To track AL1 cells for the purposes of quantifying AxFUCCI fluorescence. |

## Computational methods

### Model of developing and regenerating axolotl spinal cord

We modeled the spinal cord as a densely packed row of ependymal cells. Since all the cells are assumed identical rigid spheres, the model effectively involves only one spatial dimension: the AP

axis of the spinal cord. We assumed that cells are either cycling or quiescent, where the fraction of cycling cells is the growth fraction *GF*. We considered that each cycling cell *i* located in the position $x_i$ at the time *t* proliferates with a certain random cell cycle length lognormally distributed $T_i(x_i, t)$ and has a certain age within its cell cycle, $C_i(x_i, t)$, defined as a coordinate along the cell cycle or clock ($0 \leq C_i(x_i, t) < T_i(x_i, t)$). In the initial condition, each cycling cell has a random age $C_i(x_i, t = 0) = C_i^0$ along its particular cell cycle length $T_i(x_i, t)$, where the $C_i^0$ distribution is given by $\left(\frac{\ln 2}{T_i(x_i,t)}\right) 2^{1 - \frac{2C_i^0}{T_i(x_i,t)}}$

. As time *t* goes by, each cycling cell increases its clock $C_i(x_i, t)$ deterministically until it reaches its corresponding $T_i(x_i, t)$ value. At this precise moment, the cell divides and one daughter inherits its mother's AP coordinate while the other is intercalated between the first daughter and the posterior neighboring cell. This last feature of the model is the implementation of what we earlier defined as 'cell pushing mechanism' (*Rost et al., 2016*). After cell division, the daughter cells reinitiate their clocks and $C_i(x_i, t) = 0$. This model predicts that after a time of approximately one cell cycle length mitotic events will occur along the AP axis, contributing to the growth of the spinal cord during development (*Figure 1A*).

To study the evolution of the tissue under a regenerative setup, we focused on the tissue response to an amputation modeled by simply removing the most posterior cells. We modeled the regenerative response in the remaining $N_0$ cells by assuming that amputation triggers the release of a signal, which spreads with constant velocity anteriorly over the AP axis while recruiting the ependymal cells. We assumed that cell recruitment stops at time *τ*, rendering *λ* µm of cells anterior to the amputation plane recruited and a recruitment velocity -*λ*/*τ* during the interval $0 \leq t \leq \tau$. We notated the AP position of the most anterior cell recruited by the signal as *ξ(t)*, the recruitment limit, such that *ξ(t = τ) = -λ*.

Because regenerative ependymal cells shorten G₁ and S phases (without altering G₂ and M phases) leading to an acceleration of the cell cycle (*Rodrigo Albors et al., 2015*), we assumed that the signal-induced recruitment instructs regenerating ependymal cells precisely to reduce G₁ and S phases, effectively shortening their cell cycle (*Figure 1C*). We represent here as $G1^{long}$ and $S^{long}$ ($G1^{short}$ and $S^{short}$) the length of the corresponding phases for ependymal cells of uninjured animals (regenerating animals). We notate with $T^{long}$ and $T^{short}$ to the cell cycle length of the ependymal cells in uninjured and regenerating axolotls, respectively.

Note that a cycling cell *i* whose position $x_i$ is anterior to the recruitment limit ($x_i < \xi(t)$) is not recruited at time *t* and has a cell cycle length $T_i(x_i, t)$ equal to $T^{long}$, that is, continue cycling slowly during the simulations (*Figure 1—figure supplement 1B*). In contrast, a cycling cell *i* whose position $x_i$ is posterior to the recruitment limit ($x_i \geq \xi(t)$) within the time interval $0 < t \leq \tau$ is irreversibly recruited and consequently has a cell cycle length $T_i(x_i, t)$ equal to $T^{short}$. The progeny of the recruited cells (non-recruited cells) have a cell cycle length extracted from the same lognormal distribution of $T^{short}$ ($T^{long}$) (*Figure 1C*).

We assumed that recruitment of a cell *i* located in the position $x_i$ at time *t* induces an irreversible transformation in its cell cycle coordinate $C_i(x_i, t) \rightarrow C_i'(x_i, t)$, where $C_i(x_i, t)$ and $C_i'(x_i, t)$ are the original and transformed cell cycle coordinates, respectively. This means that the cell cycle coordinates of these cells undergo an irreversible coordinate transformation, modifying their cycling according to the cell cycle phase in which they are in at the moment of recruitment, as we describe in the following subsections (*Figure 1—figure supplement 1B*).

### When the cells are in the G₁ phase at the time of recruitment

We assumed that if at the moment of amputation *t* a cell *i* would be in a cell cycle coordinate $C_i(x_i, t)$ within $0 \leq C_i(x_i, t) \leq G1^{long} - G1^{short}$, the new cell cycle coordinate is as follows (*Figure 1C*, *Figure 1—figure supplement 2*):

$$C_i'(x_i, t) = 0 \tag{1}$$

which would induce a synchronization. In contrast, if at the moment of amputation *t* a cell *i* would be in a cell cycle coordinate $C_i(x_i, t)$ within $G1^{long} - G1^{short} \leq C_i(x_i, t) \leq G1^{long}$, the new cell cycle coordinate is

$$C_i'(x_i, t) = C_i(x_i, t) - \left(G1^{long} - G1^{short}\right) \tag{2}$$

That is, these cells continue cycling as before. Taken together, the cells in G₁ become partially synchronized (*Figure 1C*, *Figure 1—figure supplement 2*).

## When the cells are in the S phase at the time of recruitment

Taking into account that in S phase all DNA must be duplicated for cell division to occur, we considered a different mechanism to model S phase shortening based on proportional mapping. The new cell cycle coordinate of this cell is proportionally mapped to the corresponding coordinate of a shortened S phase in the next simulation step. Thus, we assumed that if at the moment of amputation $t$ a cell $i$ would be in the S phase, that is, in a cell cycle coordinate $C_i(x_i, t)$ within $G1^{long} \leq C_i(x_i, t) \leq G1^{long} + S^{long}$, the transformed cell cycle coordinate key resources relative to the S phase length is invariant (*Figure 1C*, *Figure 1—figure supplement 2*):

$$\frac{C'_i(x_i,t) - G1^{short}}{S^{short}} = \frac{C_i(x_i,t) - G1^{long}}{S^{long}} \tag{3}$$

As a consequence, the transformed cell cycle coordinate is as follows:

$$C'_i(x_i, t) = \left( C_i(x_i, t) - G1^{long} \right) \frac{S^{short}}{S^{long}} + G1^{short} \tag{4}$$

## When the cells are either in the G₂ or M phase at the time of recruitment

We previously demonstrated that the sum of G₂ and M phase lengths of the ependymal cells of axolotl spinal cords was conserved after amputation (*Rodrigo Albors et al., 2015*). Hence, once a cell $i$ is in the joint G₂+ M phases, the remaining time to complete the cell cycle is the same for both, the original and the transformed cell cycle coordinates (*Figure 1C*):

$$T^{long} - C_i(x_i, t) = T^{short} - C'_i(x_i, t) \tag{5}$$

As a consequence, the transformed cell cycle coordinate can be calculated as follows:

$$C'_i(x_i, t) = C_i(x_i, t) + \left( T_C^{short} - T_C^{long} \right) \tag{6}$$

## When the cells are in the G₀ phase at the time of recruitment

We assumed that if a recruited cell $i$ is quiescent at the moment $t$ of recruitment, that is, in G₀, it progress from this phase to the short-G₁ phase after a certain delay $t_{G0-G1}$, which was fixed to reproduce the growth fraction kinetics previously reported (Figure 3B in *Rost et al., 2016*):

$$C'_i(x_i, t + t_{G0-G1}) = 0 \tag{7}$$

## Model parametrization

The model parameters are summarized in *Table 1*. Briefly, the ependymal cell length along the AP axis, the distributions of cell phases durations and growth fraction were fixed from our previous publication (*Rodrigo Albors et al., 2015*). The only free model parameters are the remaining anterior cells after amputation $N_0$, the maximal length $\lambda$ along the AP axis of the putative signal and $\tau$, the maximal time of cell recruitment.

## Fitting procedure of the experimental switchpoint with the theoretical recruitment limit ξ(t)

The experimentally obtained switchpoint of the regenerating axolotl spinal cord (extracted from *Rost et al., 2016*) was fitted with the model-predicted recruitment limit $\xi(t)$. We followed an ABC method to estimate the distribution of the parameters that better reproduces the experimental switchpoint data by our recruitment limit model. The ABC methods bypass the requirement for evaluating likelihood functions and captures the uncertainty in estimates of model parameters (*Csilléry et al., 2010*). In particular, we used pyABC (*Klinger et al., 2018*), a high-performance framework implementing

a Sequential Monte Carlo scheme (ABC-SMC), which provides a particularly efficient technique for Bayesian posterior estimation (*Toni et al., 2009*).

Briefly, we generated a series of stochastic simulations from the model (described in the previous section) from sampled points of the parameter space. Each fitting run was initialized with a population size of 1000 samples. All the prior distributions of the parameters were defined as a discrete uniform distribution: $N_0 \sim$ unif{100, 300}, $\lambda \sim$ unif{500, 1500}, and $\tau \sim$ unif{1,192}, where the limits of $\tau$ were basically given by the entire experimental observation time (8 days) in hours. The limits of $\lambda$ (in μm) and $N_0$ were initially estimated by previous simulation trials.

The sampled parameter values were accepted only when the distance function $d$ between simulated recruitment limit and experimental switchpoint was lower than a given tolerance $\varepsilon$. The distance function was defined as follows:

$$d = \sqrt{\frac{\sum_i (x_i - \mu_i)^2}{\sigma_i^2}},$$

(8)

where $\mu_i$, $\sigma_i$, and $x_i$ correspond to the mean, standard deviation of the experimental switchpoint, and the simulated recruitment limit, respectively, at the experimental time points $i$ (4, 6, and 8 days). At each iteration, the parameter distributions were updated and re-sampled. The new tolerance $\varepsilon$ was then calculated as the median of the distances from the last accepted sample population. The outcome of the algorithm was a sample of parameter values inferring their posteriors distributions (*Figure 2—figure supplement 1A,B*). Convergence of the method was assessed by following the value of $\varepsilon$ and the acceptance rate, defined as the accepted number of simulations divided by the total number of simulations at each iteration step (*Figure 2—figure supplement 1C,D*).

## Clones trajectories and velocities

We calculated the clone trajectories following the positions of each clone in random simulations. When a cell divided, we kept the mean position of the clone cells as the clone position. In *Figure 2—figure supplement 2*, a total of 11 tracks are shown, the first trajectory starts at 0 (the amputation plane) and the last at –1100 μm (with a sampling of 50 μm, approximately). To estimate the mean velocity of clones at different spatial positions in this model, the space along the AP axis was subdivided into 800 μm bins. For each clone trajectory, the positions were grouped according to these bins. Groups containing less than two measurements were excluded. The average clone velocity for each group was estimated with linear regression. Then, the mean and standard deviation of the velocity of all the clones in a bin were calculated.

## Coordinate system

In all our simulations, the time starts with the event of amputation. Space corresponds to the AP axis, where 0 represents the amputation plane and positive (negative) values are posterior (anterior) locations.

## Model implementation and computational tools

The models were implemented in Python 3.0. Simulations and data analysis were performed using Numpy (*Oliphant, 2006*) and Pandas (*McKinney, 2010*) while data visualization was executed with Matplotlib (*Hunter, 2007*).

## Supplementary notebooks

Jupyter Notebook (http://jupyter.org/) containing the source code for all computations performed and referred to as *Cura Costa et al., 2021* in this study can be found at https://doi.org/10.5281/zenodo.4557840.

## **Experimental materials and methods**

## Molecular biology

AxFUCCI plasmid was constructed by standard restriction cloning. Relevant features of the AxFUCCI plasmid are (i) SceI site for stable transgenesis; (ii) CAGGs synthetic promoter for ubiquitous expression; (iii) $G_0/G_1$-AxFUCCI probe (axolotl Cdt1$_{[aa1-128]}$-GSAGSAAGSGEF glycine/serine linker-mVenus);

(iv) T2A 'self-cleaving' viral peptide; (v) S/G$_2$-AxFUCCI probe (mCherry-SGGGGGSGGGGS glycine/serine linker-axolotl Gmnn$_{[aa1-93]}$); (vi) rabbit beta-globin polyadenylation sequence; and (vii) SceI site for stable transgenesis. PCR products were amplified using the primers listed in the Key resources table and ligated in two rounds into a vector already harboring SceI sites, a CAGGs promoter, a T2A sequence followed by mCherry and rabbit beta-globlin polyadenylation sequence.

Primers were purchased as 20 µM stocks (Sigma-Aldrich, standard de-salt). AxFUCCI plasmid sequence was verified by Sanger sequencing.

## AL1 cell culture
The immortalized axolotl 'AL1' cell line was grown in a humidified incubator at 25°C, with 2% CO$_2$. The cell culture medium contains 62.5% MEM, 10% fetal bovine serum, 25% water, supplemented with 100 U penicillin-streptomycin, glutamine, insulin. AL1 cells were passaged every week at a ratio of 1:2 into gelatin-coated flasks.

## AL1 cell electroporation and live imaging
Electroporation was performed using a Neon Transfection System (Thermo Fisher Scientific). 50,000 AL1 cells were electroporated with 1 µg of AxFUCCI plasmid in 70% PBS/water using the following settings: 750 V, 35 ms pulse width, 3 pulses. Electroporated AL1 cells were plated onto a glass-bottomed Ibidi imaging dish coated with gelatin. After 2 days, the cell culture medium was exchanged, and the dish placed in a Celldiscoverer 7 automated live-cell imaging microscope chamber (Zeiss). The microscope chamber was maintained at 25°C, with 2% CO$_2$. Cells were imaged hourly over the course of 7 days for Venus and mCherry fluorescence, and brightfield.

## Fluorescence intensity track measurements
AxFUCCI fluorescence intensities were measured using the TrackMate plugin for Fiji (*Tinevez et al., 2017*). Fluorescence intensities were normalized to the maximum intensity observed for the respective fluorophore during the experiment.

## DNA quantification by flow cytometry
AxFUCCI-electroporated AL1 cells were incubated for 90 min with cell culture medium containing 10 µg/ml Hoechst DNA stain. After incubation, AL1 cells were washed once with 70% PBS/water, then dissociated into single cells using Trypsin. Dissociation was terminated by adding a 1:1 volume of serum-containing cell culture medium. Cells were pelleted and re-suspended in 70% PBS/water, then filtered through a 50 µm cell filter. Cells were analyzed for DNA content using a BD LSRFortessa Flow Cytometer and FlowJo software.

## Axolotl husbandry and transgenesis
*d/d* and AxFUCCI axolotls (*A. mexicanum*), snout-to-tail length 5.5 cm, were raised in individual aquaria. Axolotl breedings were performed by the IMP animal facility. All experiments were performed in accordance with locally applicable ethics committee guidelines and within a framework agreed with the Magistrate of Vienna (Genetically Modified Organism Office and MA58, City of Vienna, Austria). Axolotls were anesthetized with benzocaine (Sigma) diluted in tap water prior to amputation and/or imaging.

AxFUCCI axolotls were generated by I-SceI meganuclease-mediated transgenesis using previously described methods (*Sobkow et al., 2006*). Briefly, one-cell-stage fertilized *d/d* axolotl eggs were de-jellied and injected with 5 nl of injection mix (~0.5 ng AxFUCCI plasmid and 0.005 U I-SceI meganuclease [NEB] diluted in 1× CutSmart buffer [NEB]). Injected axolotl eggs were maintained in 0.1× MMR/tap water at room temperature until screening. Transgenic founder (F0) animals were identified by their Venus fluorescence using an AXIOzoom V16 widefield microscope (Zeiss). F1 germline-transmitted AxFUCCI animals were used for all experiments in this study. Sample sizes were determined empirically and within the confines of experimentation under COVID19 pandemic-induced operational restrictions. Requests for AxFUCCI axolotls should be directed to, and will be fulfilled by, EMT (elly.tanaka@imp.ac.at). The designation for AxFUCCI axolotls is tgSceI(*CAGGs:CDT1[aa1-128]-mVenus-T2A-mCherry-GMNN[aa1-93]*)[Etnka], according to the standardized nomenclature proposed by *Nowoshilow et al., 2021*.

### Tissue clearing and lightsheet imaging

AxFUCCI tails were fixed overnight at 4°C in 4% paraformaldehyde. Fixed tails were washed well with PBS, then de-lipidated for 30 min at 37°C in Solution-1 of the DEEP-clear tissue clearing protocol (10% v/v THEED, 5% v/v Triton X-100, 25% w/v urea in water) (*Pende et al., 2020*). De-lipidated tails were washed with PBS, then incubated for 2 hours in PBS containing 10 µg/ml DAPI. Tails were washed well, then incubated overnight in Easyindex refractive index matching solution (LifeCanvas Technologies). Samples were kept dark at all times to prevent bleaching of AxFUCCI fluorescence. Cleared AxFUCCI tails were imaged in EasyIndex solution using a LightSheet.Z1 microscope (Zeiss) and custom chamber.

### Preparation of tissue sections

AxFUCCI tails were fixed overnight at 4°C in 4% paraformaldehyde. Fixed tails were washed well with PBS, then incubated overnight in 30% sucrose in PBS. The following day samples were embedded in optimal cutting temperature (OTC) compound, frozen on dry ice, and stored at –80°C until sectioning. Cryosections of 10 µm thickness were prepared from frozen blocks and stored at –20°C until use.

### Immunostaining and imaging of tissue sections

Cryosections were warmed up to room temperature, then washed extensively with PBS to remove OCT. Sections were blocked for 2 hours at room temperature with 10% normal goat serum (NGS) diluted in PBS containing 0.2% Triton X-100 (PBTx). Blocked samples were incubated with primary antibody diluted in 1% NGS overnight at 4°C. The following day sections were washed well with PBTx, then stained with Alexa Fluor-conjugated secondary antibodies diluted 1:500 in PBTx for 2 hours at room temperature. DAPI was included in the secondary staining solution at a concentration of 10 µg/ml. Sections were washed well with PBTx and mounted in Mowiol containing 2.5% DABCO (Sigma). The following primary antibodies were used in this study: anti-NeuN (Millipore MAB377, mouse, 1:500) and anti-Sox2 (rabbit, 1:1000, *Fei et al., 2016*). Images were acquired using a LSM980 AxioObserver inverted confocal microscope (Zeiss).

### EdU administration and detection

Anesthetized axolotls were injected intraperitoneally with 400 µM EdU (diluted in PBS) at a dosage of 20 µl/g. FastGreen dye (Sigma-Aldrich) was added to the injection mix to aid visualization. Injected axolotls were kept out of water for a 20 min recovery period under benzocaine-soaked towels. After recovery, injected axolotls were returned to water. Following the desired pulse-chase period, axolotls were sacrificed, tail tissue fixed, and cryosections prepared.

EdU detection was performed using the Click-iT 647 EdU detection kit (Thermo Fisher Scientific) according to the manufacturer's instructions.

### Image analysis

Lightsheet data (AxFUCCI axolotl tails) were digitally re-sectioned or rendered in 3D using Imaris software (Oxford Instruments). For quantification of wholemount data, spinal cords were digitally re-sectioned longitudinally to yield a continuous strip of spinal cord lumen, and images were exported as TIFFs for cell counting in Fiji. Ependymal cell cell cycle phases were quantified from 25-µm-thick digital sections. Ependymal cells were defined as cells in direct contact with the spinal cord lumen. Spinal cord outgrowth was calculated relative to the amputation plane by taking the severed notochord as an indicator for the amputation plane (0 µm) and measuring between this point and the most posterior tip of the spinal cord at that time point.

Celldiscoverer 7 videos (AL1 cells) were cropped using ZEN blue software (Zeiss), then analyzed using the TrackMate plugin for Fiji, as described above. Tissue section images were analyzed and quantified using Fiji software (*Schindelin et al., 2012*).

### Determining the AP border between two adjacent spatial zones within axolotl regenerating spinal cords

We tested whether ependymal cells in the different cell cycle phases would be heterogeneously distributed along the AP axis of the regenerating spinal cord. To that aim, we fitted the experimental

spatial AP profiles of the percentages of $G_0/G_1$-AxFUCCI- and $S/G_2$-AxFUCCI-expressing cells, per animal, with a mathematical model assuming two adjacent homogeneous spatial zones separated by an AP border, as follows:

$$g0g1(x) = \begin{cases} g0g1_a & if\ x < \text{AP border} \\ g0g1_p & if\ x \geq \text{AP border} \end{cases}$$

(9)

$$sg2(x) = \begin{cases} sg2_a & if\ x < \text{AP border} \\ sg2_p & if\ x \geq \text{AP border} \end{cases}$$

(10)

where $g0g1(x)$ and $sg2(x)$ are the model variables describing the spatial distribution of $G_0/G_1$ and $S/G_2$ cells along $x$, the spatial position along the AP axis. The model parameters are $g0g1_a$, the anterior percentage of $G_0/G_1$-AxFUCCI-expressing cells; $g0g1_p$, the posterior percentage of $G_0/G_1$-AxFUCCI-expressing cells; $sg2_a$, the anterior percentage of $S/G_2$-AxFUCCI-expressing cells; $sg2_p$, the posterior percentage of $S/G_2$-AxFUCCI expressing cells; and *AP border*, the border between the anterior and the posterior zones, assumed equal for $G_0/G_1$-AxFUCCI and $S/G_2$-AxFUCCI cells.

We fitted the model simultaneously to the AP profile of the percentage of $G_0/G_1$-AxFUCCI- and $S/G_2$-AxFUCCI-expressing cells of each animal and each time by using an ABC method (see Section 1.3 for more details of the computational implementation). Each fitting was initialized with a constant population size of 1000 samples. The parameters priors were defined as a discrete uniform between 0% and 100% for $g0g1_a$, $g0g1_p$, $sg2_a$, and $sg2_p$. The prior for the *AP border* was also a discrete uniform covering all the measured positions along the AP axis. The distance function between the experimental FUCCI data and the two-zones model was defined as

$$d = \sum_x \sqrt{\left(G0/G1_{exp}(x) - g0g1(x)\right)^2 + \left(S/G2_{exp}(x) - sg2(x)\right)^2}$$

(11)

where $G0/G1_{exp}(x)$ and $S/G2_{exp}(x)$ are the percentages of $G_0/G_1$-AxFUCCI- and $S/G_2$-AxFUCCI-expressing cells, respectively, determined at the bin $x$ of the spinal cord AP axis (for each animal and each time).

Each fitting procedure had a total of 30 iterations. Fitting results are shown in *Figure 4B*, *Figure 4—figure supplement 1*, and *Figure 4—figure supplement 2*.

For each time, we compared the anterior versus the posterior zones of $G_0/G_1$-AxFUCCI and $S/G_2$-AxFUCCI data simultaneously by performing a Kolmogorov–Smirnov test between the best-fitting parameters $g0g1_a$, $g0g1_p$ versus $sg2_a$, $sg2_p$. Although anterior and posterior zones were indistinguishable from 0 to 3 days post-amputation, we found a significant difference between anterior and posterior zones in the $G_0/G_1$-AxFUCCI and $S/G_2$-AxFUCCI data at days 4 and 5 (*Figure 4—figure supplement 1*). The best-fitting AP border detected for each time post-amputation are in *Figure 4B* as vertical gray areas at days 4 and 5, post-amputation.

## Statistical analysis and data representation

In *Figure 4*, Kolmogorov–Smirnov test was implemented by using the Scipy (*Virtanen and SciPy 1.0 Contributors, 2020*) library. Numpy's (*Harris et al., 2020*) high-level mathematical functions were used all along the simulations and data analysis. *Figure 2*, *Figure 2—figure supplement 2*, *Figure 2—figure supplement 3*, *Figure 2—figure supplement 4*, and *Figure 3—figure supplement 4B* were made using Matplotlib (*Hunter, 2007*) while *Figure 4B*, *Figure 4—figure supplement 1*, *Figure 4—figure supplement 2*, *Figure 6*, *Figure 6—figure supplement 1*, and *Figure 6—figure supplement 2* were performed with Seaborn (*Waskom et al., 2012*). In *Figure 5*, *Figure 5—figure supplement 1*, and *Figure 3—figure supplement 4*, statistical analyses were performed using R. AxFUCCI data were tested for assumptions of normality (Shapiro–Wilk test) and equality of variance (Levene's test) in order to determine the appropriate statistical tests to perform. No data were excluded. Details of statistical tests and their outcomes can be found in the relevant figure legends. Statistical significance was defined as $p < 0.05$. These graphs were plotted using Prism (GraphPad). All the figures were compiled in Adobe Illustrator.

## AxFUCCI axolotl availability

The designation for AxFUCCI axolotls is tgSceI(*CAGGs:CDT1[aa1-128]-mVenus-T2A-mCherry-GMNN[aa1-93]*)[Etnka], according to the standardized nomenclature proposed by *Nowoshilow et al., 2021*. We envision that AxFUCCI axolotls will serve as useful tools for the community, and these transgenic animals are freely available upon request from EMT (elly.tanaka@imp.ac.at).

## Acknowledgements

We thank Fabian Rost for critical comments on the manuscript. We also thank the members of the Chara lab and especially Alberto Ceccarelli for interesting discussions. We are grateful to Pietro Tardivo (tissue clearing and imaging), Anastasia Polikarpova (flow cytometry), Alberto Moreno-Cencerrado and Pawel Pasierbek (BioOptics facility, Vienna Biocenter), and the animal caretaker team (Vienna Biocenter), whose support enabled experiments to be performed during the COVID-19 pandemic.

## Additional information

### Funding

| Funder | Grant reference number | Author |
| --- | --- | --- |
| Agencia Nacional de Promoción Científica y Tecnológica | PICT 2014-3469 | Osvaldo Chara |
| Agencia Nacional de Promoción Científica y Tecnológica | PICT 2017-2307 | Osvaldo Chara |
| Agencia Nacional de Promoción Científica y Tecnológica | PICT-2019-2019-03828 | Osvaldo Chara |
| Consejo Nacional de Investigaciones Científicas y Técnicas | Doctoral Student Fellowship | Emanuel Cura Costa |
| European Research Council | Advanced Grant 742046 | Elly M Tanaka |
| Human Frontier Science Program | fellowship LT000785/2019-L. | Leo Otsuki |
| Horizon 2020 - Research and Innovation Framework Programme | Marie Skłodowska-Curie grant agreement No 753812 | Aida Rodrigo Albors |
| Austrian Science Fund | SFB-F78 | Elly M Tanaka |

The funders had no role in study design, data collection and interpretation, or the decision to submit the work for publication.

### Author contributions

Emanuel Cura Costa, Data curation, Investigation, Methodology, Software, Validation, Visualization; Leo Otsuki, Data curation, Investigation, Methodology, Validation, Visualization, writing-review-and-editing; Aida Rodrigo Albors, conceptualization, Investigation, writing-original-draft, writing-review-and-editing; Elly M Tanaka, conceptualization, writing-original-draft, writing-review-and-editing; Osvaldo Chara, conceptualization, funding-acquisition, Investigation, project-administration, Software, supervision, Validation, writing-original-draft, writing-review-and-editing

### Author ORCIDs

Emanuel Cura Costa http://orcid.org/0000-0002-7030-2077
Leo Otsuki http://orcid.org/0000-0001-6107-2508
Aida Rodrigo Albors http://orcid.org/0000-0002-9573-2639
Elly M Tanaka http://orcid.org/0000-0003-4240-2158
Osvaldo Chara http://orcid.org/0000-0002-0868-2507

## Ethics

The licenses necessary for work with genetically modified organisms (GMOs) and for experiments specifically involving axolotls (Ambystoma mexicanum) were obtained from the relevant authorities and have been implemented at the IMP in accordance with applicable international, EU and national (Austrian) guidelines. The license for work with GMOs at safety levels 1 and 2 was approved by the GMO office of the Austrian authorities and was issued on 16/03/2017 with no end date (BMGF-76110/0017-II/B/16c/2017).The axolotl research license numbers are GZ: 51072/2019/16 (valid 09/05/2019 - 28/02/2024) and GZ: MA58/665226/2019/21 (valid 24.02.2020 - 30.09.2024). These are approved by the City of Vienna, MA58.There is a dedicated veterinarian for the animal facility, as well as an animal welfare consultant. Animal facility inspections are performed yearly by the City of Vienna. MA58 and the animal licenses and animal husbandry conditions are updated in dialogue with animal welfare authorities. Axolotls (Ambystoma mexicanum) were raised in individual aquaria. Axolotl breedings were performed by the IMP animal facility. All experiments were performed in accordance with locally applicable ethics committee guidelines and within a framework agreed with the Magistrate of Vienna (Austria). Axolotls were anesthetized with benzocaine (Sigma) diluted in tap water prior to amputation and/or imaging, to minimize suffering.

## Decision letter and Author response

Decision letter https://doi.org/10.7554/eLife.55665.sa1
Author response https://doi.org/10.7554/eLife.55665.sa2

# Additional files

## Supplementary files

• Transparent reporting form

## Data availability

Jupyter Notebook (http://jupyter.org/) containing the source code for all computations performed and referred to as Cura Costa, Otsuki et al., 2021 in this study can be found at https://doi.org/10.5281/zenodo.4557840.

The following dataset was generated:

| Author(s) | Year | Dataset title | Dataset URL | Database and Identifier |
|---|---|---|---|---|
| Cura Costa E, Otsuki L, Rodrigo Albors A, Tanaka EM, Chara O | 2021 | | https://doi.org/10.5281/zenodo.4557840 | Zenodo, 10.5281/zenodo.4557840 |

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
