## [Decision Letter]

**Acceptance summary:**

By coupling modeling with experimental data using FUCCI technology to visualize cell cycle dynamics in vivo, the authors demonstrate that spinal cord regeneration in the axolotl is likely to occur in response to a signal that recruits synchronously cycling ependymal cells shortly after injury. These data help elucidate important aspects of spinal cord regeneration.

**Decision letter after peer review:**

Thank you for submitting your article "Modeling the spatiotemporal control of cell cycle acceleration during axolotl spinal cord regeneration" for consideration by *eLife*. Your article has been reviewed by 2 peer reviewers, and the evaluation has been overseen by Alejandro Sánchez Alvarado as the Reviewing Editor and Marianne Bronner as the Senior Editor. The following individual involved in the review of your submission has agreed to reveal their identity: Carsten Marr (Reviewer #1).

The reviewers have discussed the reviews with one another and the Reviewing Editor has drafted this decision to help you prepare a revised submission.

As the editors have judged that your manuscript is of interest, but as described below that additional experiments are required before it is published, we would like to draw your attention to changes in our revision policy that we have made in response to COVID-19 (https://elifesciences.org/articles/57162). First, because many researchers have temporarily lost access to the labs, we will give authors as much time as they need to submit revised manuscripts. We are also offering, if you choose, to post the manuscript to bioRxiv (if it is not already there) along with this decision letter and a formal designation that the manuscript is 'in revision at *eLife*'. Please let us know if you would like to pursue this option. (If your work is more suitable for medRxiv, you will need to post the preprint yourself, as the mechanisms for us to do so are still in development.)

Summary:

E.Cura Costa et al. present a spatio-temporal modeling approach to describe cell cycle acceleration during Axolotl spinal cord regeneration. The model is based on previous in vivo observations and reveals spatial and temporal intracellular coordination with an optimal parameter set. The model is calibrated with experimental data and predicts that a signal that comes into play 24 hours post-amputation and recruits cells located within one millimeter anterior to the injury site could explain the spatiotemporal pattern of cell proliferation after injury. The model further allows assessing the individual contributions of S and G1 phase shortening required to explain the experimentally observed outgrowth dynamics. The idea and necessity of this model are nicely motivated by previous in vivo findings and are explained understandably along with the figures. The mathematical predictions could help to identify possible signals in spinal cord regeneration in the future. However, we think that the evidence for this model hypothesis is not convincing. In particular, other model hypotheses are not quantitatively rejected. In its current form, it simply says that a particular hypothesis about G1 and S phase changes fits the data, but many other models would probably do so as well. Experimental validation of the assumptions made and predictions of the model would make the paper stronger.

Essential revisions:

1. Model comparisons should be more rigorous.

– The comparison of different hypotheses in Figure 4 is only qualitative. No proper parameter fitting is involved.

– Parameter fitting should also consider the variance (not only the mean) of experimental measurements.

– We are not convinced that the outgrowth dynamics suffices to compare the different G1/S shortening hypotheses. Can you prove that these parameters are indeed identifiable?

2. The mathematical model is based on simplifying assumptions, i.e. the spinal cord is modeled as a row of rigid spheres representing the cells. Here, the question arises whether the quantitative values provided are indeed representative of the in vivo situation and will, therefore, have a significant impact on future studies on the mechanisms of spinal cord regeneration. So far, the existence of the proposed signal has only been evaluated through mathematical predictions but has not been confirmed experimentally.

3. How likely is the fact that a single signal as mathematically introduced here is actually the driving force for spinal cord regeneration? Couldn't it also be a combination of different factors?

4. It should be better emphasized which experimental data were used for fitting and which were used to validate the model. For instance, it seems like the experimental switchpoint data were used for fitting, while the agreement with experimental outgrowth in *Sox2* knock-out axolotls can be regarded as a validation. Emphasizing how the model draws together different data sets will highlight the predictive power of the model and enhance the reader's trust in the model predictions.

5. The model assumptions are not always well-motivated.

– Why is the cell cycle normally distributed, giving in principle also rise to negative values? A log-normal or lag-exponential dist. would be more intuitive.

– Is 'we assumed that recruited cells whose cell cycle coordinates belong to G2 or M when t = τ will continue cycling as before' substantiated by the data? Why shouldn't all cells in the zone change their cell cycle? Or a random fraction? Can we rule out these possibilities with model comparison?

6. The optimal set of parameters is obtained through a brute-force sampling approach over a certain parameter domain and subsequent evaluation of least-squares. How sensitive are results with respect to the different parameters? How were the parameter domains and corresponding step sizes chosen? Without this information, it is difficult to judge the robustness of the results.

7. Predictions should be experimentally verified.

– Why not show the 1mm zone with e.g. BrdU staining (experiments seems to have been done already in Rodrigo Albors et al., 2015) or AxMLP staining?

– Is there evidence in the data that substantiates 'partial synchronization of cells transiting through G1'? This would be a strong indication that the assumed G1 mechanism is indeed in place.

---

## [Author Response]

Essential revisions:1. Model comparisons should be more rigorous.– The comparison of different hypotheses in Figure 4 is only qualitative. No proper parameter fitting is involved.

It seems that we were not clear regarding when we fitted the model and when we did predictions. Thanks to the reviewers’ comments, we clarified these points in the revised version of the manuscript.

Essentially, we fitted the modelled recruitment limit to the experimental switchpoint previously reported (Figure 2A of the revised manuscript). By fixing the parameter in their best-fitting values, we predicted the outgrowth and observed a strong result: this prediction is in agreement with the experimentally observed spinal cord outgrowth in the axolotl (Figure 2B of the revised manuscript). We then would like to remark that the simulated outgrowth were not fitted to the experimental data, but predictions of the model.

The model assumes that both S and G1 phases are reduced. In the former Figure 4 (and current Figure 2D of the revised version) we compared this prediction with the case in which there is no reduction and with two hypothetical scenarios in which either S or G1 are reduced. Unfortunately, these two scenarios cannot be experimentally evaluated. However, we compared the outgrowth prediction in the absence of any cell cycle phase reduction with an independent experimental dataset in Figure 2C of the revised manuscript.

– Parameter fitting should also consider the variance (not only the mean) of experimental measurements.

We thank the reviewer for the comment. We included the variance of the experimental switchpoint curve in the fitting of the model in the revised manuscript.

– We are not convinced that the outgrowth dynamics suffices to compare the different G1/S shortening hypotheses. Can you prove that these parameters are indeed identifiable?

The effect of S shortening in the outgrowth can be identified by considering the length of the long and short S, the very short G2+M phases, the time of observation (8 days) and the proximity of S to M phase, after which the cell divides and the tissue grows. Taking these into account, the effect of S shortening in the outgrowth is indeed detectable and we now provide these details when presenting these results (Figure 4 of the previous manuscript and Figure 2 D of the revised manuscript).

More importantly, we generated FUCCI axolotls that allowed us to classify cells based on their cell cycle phase during spinal cord regeneration and test our hypothesis (Figure 3, 4, 5 and 6 A,B of the revised manuscript).

2. The mathematical model is based on simplifying assumptions, i.e. the spinal cord is modeled as a row of rigid spheres representing the cells. Here, the question arises whether the quantitative values provided are indeed representative of the in vivo situation and will, therefore, have a significant impact on future studies on the mechanisms of spinal cord regeneration. So far, the existence of the proposed signal has only been evaluated through mathematical predictions but has not been confirmed experimentally.

This is an important point and we thank the reviewer for bringing it up. All the assumptions simplify aspects of the in vivo situation, as it is expected from a modelling approach. We developed each model assumption in the spirit of the Occam’s razor: we tried to conceive simple assumptions (ideally the simplest assumptions) while still satisfying experimental data. The particular assumption mentioned by the reviewer is an example of how we followed this parsimonious procedure. Although the model assumption of cells being rigid spheres of identical diameter is indeed very simple, it was motivated by our in vivo experimental data. In particular, we determined that cell density along AP axis was constant during regeneration and we were able to measure the length of ependymal cells along this AP axis with a small coefficient of variation (36). The comment of the reviewer prompted us to test whether cells compressing within their shortest and longest possible lengths along the AP axis (based on experimental data) would affect the predicted outgrowth. (Figure 2—figure supplement 3A,B of the revised version of the manuscript). We also tested whether cells with non-uniform lengths along the AP axis (but assuming a normal distribution, consistent with the experimental data) would impact on the predicted outgrowth (Figure 2—figure supplement 3C of the revised version of the manuscript). Our new results indicate that outgrowth predictions are not sensitive to these variations in cell length. These results are presented as the Figure 2—figure supplement 3A-C of the current version of the manuscript.

More importantly, in the previous version of the model, we conceived a signal that would instantaneously recruit all cells within certain domain of the AP axis after certain delay. In the light of the reviewer’s question, we decided to reduce the simplicity of the model and assumed that the signal starts gradually recruiting cells right after amputation. This assumption turned out to be more consistent with the new in vivo data using AxFUCCI which indicates that cell recruitment is not an all-or-none event but rather a progressive response.

We modified other two assumptions, reducing also their simplicity. These two assumptions describe the main stochastic elements of the model: the distribution of the cell cycle length and the initial distribution of cell age along the cell cycle. We describe this in the answer to the question 5.

3. How likely is the fact that a single signal as mathematically introduced here is actually the driving force for spinal cord regeneration? Couldn't it also be a combination of different factors?

We conceived the signal introduced in our study as a general concept. In the previous version of the manuscript, we speculated that the signal could be of a chemical, electrical or mechanical nature. As the reviewer mentioned, we cannot rule out the possibility that the signal could be a combination of individual signals. For instance, amputation could trigger a morphogen, which in turn activate a second morphogen, which recruits resident ependymal cells. What our study suggests is that, whatever signals are at work to drive regeneration of the spinal cord, they individually or collectively operate with the spatiotemporal distribution that we report. In a way, our signal could be considered as an “equivalent signal” to a set of unknown signals, analogous to the equivalent resistance in an electrical circuit of an arbitrary number of resistances. Following the reviewer suggestion, we now acknowledge the combination of factors in the discussion of the revised manuscript.

4. It should be better emphasized which experimental data were used for fitting and which were used to validate the model. For instance, it seems like the experimental switchpoint data were used for fitting, while the agreement with experimental outgrowth in Sox2 knock-out axolotls can be regarded as a validation. Emphasizing how the model draws together different data sets will highlight the predictive power of the model and enhance the reader's trust in the model predictions.

We thank the reviewer for the suggestion. In the revised manuscript, we emphasized when we fitted the model to certain experimental data and when we made predictions with the model and compare those predictions with independent experimental data. As mentioned above, we fitted the model to the previously reported experimental switchpoint curve. By using the best-fitting parameter values of the model, we predicted the simulated outgrowth in the presence and in the absence of recruitment and compared these predictions with the spinal cord outgrowth observed in normal conditions (current Figure 2 B of the revised version of the manuscript) and when knocking out *Sox2* (current Figure 2 C of the revised version of the manuscript).

5. The model assumptions are not always well-motivated.– Why is the cell cycle normally distributed, giving in principle also rise to negative values? A log-normal or lag-exponential dist. would be more intuitive.

We thank the reviewer for raising these points. Although with the parametrization of the normal distribution negative values were not actually observed, the reviewer is right and a lognormal distribution would be a more correct choice. Following the advice, we used this distribution in the current version of the model. As expected due to the parametrization of the distribution, the results did not change much. More importantly, in our previous version of the model we assumed that at the moment of amputation (the initial condition), a cell would be at a random position within the cell cycle following a uniform distribution. We improved this assumption by assuming an exponential distribution in the revised manuscript.

– Is 'we assumed that recruited cells whose cell cycle coordinates belong to G2 or M when t = τ will continue cycling as before' substantiated by the data? Why shouldn't all cells in the zone change their cell cycle? Or a random fraction? Can we rule out these possibilities with model comparison?

This assumption is indeed well motivated by our previous data (Rodrigo Albors et al., 2015): we previously observed that the length of G2+M is 9 hours in non-regenerating and regenerating ependymal cells. Moreover, 9 hours is basically negligible compared to the shortest phase in regenerating conditions. Our assumption is not only the simplest but also the only one that satisfies these two facts.

6. The optimal set of parameters is obtained through a brute-force sampling approach over a certain parameter domain and subsequent evaluation of least-squares. How sensitive are results with respect to the different parameters? How were the parameter domains and corresponding step sizes chosen? Without this information, it is difficult to judge the robustness of the results.

We thank the reviewer for the comment. We replaced the fitting procedure that we followed in the first version of the manuscript by an inference Approximate Bayesian Computation method, now described in Supplementary methods section 1.3. We showed the sensitivity of the results to the different parameters by depicting the posterior distributions (Figure 2 – supplementary figure 1 of the revised version of the manuscript) and the effect in the predicted outgrowth when moving away from the best-fitting values in the parameters space (Figure 2 – supplementary figure 4 of the revised version of the manuscript).

7. Predictions should be experimentally verified.– Why not show the 1mm zone with e.g. BrdU staining (experiments seems to have been done already in Rodrigo Albors et al., 2015) or AxMLP staining?

For the revised manuscript we generated AxFUCCI axolotls to visualize the recruitment zone (Figure 3). We found that both G0/G1-AxFUCCI and S/G2-AxFUCCI fluorescence could be used to measure recruitment zone size and that, importantly, this experimentally-determined size matched the theoretical one predicted by our model (Figure 4, Figure 6).

– Is there evidence in the data that substantiates 'partial synchronization of cells transiting through G1'? This would be a strong indication that the assumed G1 mechanism is indeed in place.

Cell cycle synchronization has long been difficult to demonstrate experimentally. Nevertheless, we succeeded in addressing this issue by capitalizing on FUCCI technology. AxFUCCI axolotls enabled us to sensitively distinguish cell cycle phases in ependymal cells after amputation. In addition to broadly labelling G0/G1 phase versus S/G2 phase, AxFUCCI labelled short cell cycle transitions (G1>S and G2>M), which could be used as discrete landmarks in the cell cycle against which we could check for partial cell cycle synchronization (Figure 3). Labelling of both cell cycle transitions is not observed in other FUCCI models, and appears to be a fortuitous property of AxFUCCI. Our day-by-day quantifications revealed a significant (4.5-fold) increase in G1>S transition ependymal cells in the first and second days post-amputation compared to baseline levels (Figure 5). Moreover, we found a high degree of synchrony in cell cycle phase progression at the population level (day 0: largely G0/G1, day 1 and 2: G1/S, day 3 onwards: S, day 4 onwards: M). These two observations are strong support for partial synchronization of cells transiting through G1 after amputation.